# missForestPredict—Missing data imputation for prediction settings

**Elena Albu**[1], **Shan Gao**[1], **Laure Wynants**[1,2,3], **Ben Van Calster**[1,2]*

**1** Department of Development & Regeneration, KU Leuven, Leuven, Belgium, **2** Leuven Unit for Health Technology Assessment Research (LUHTAR), KU Leuven, Leuven, Belgium, **3** School for Public Health and Primary Care, Maastricht University, Maastricht, The Netherlands

* ben.vancalster@kuleuven.be

## Abstract

Prediction models are used to predict an outcome based on input variables. Missing data in input variables often occur at model development and at prediction time. The missForestPredict R package proposes an adaptation of the missForest imputation algorithm that is fast, user-friendly and tailored for prediction settings. The algorithm iteratively imputes variables using random forests until a convergence criterion, unified for continuous and categorical variables, is met. The imputation models are saved for each variable and iteration and can be applied later to new observations at prediction time. The missForestPredict package offers extended error monitoring, control over variables used in the imputation and custom initialization. This allows users to tailor the imputation to their specific needs. The missForestPredict algorithm is compared to mean/mode imputation, linear regression imputation, mice, k-nearest neighbours, bagging, miceRanger and IterativeImputer on eight simulated datasets with simulated missingness (48 scenarios) and eight large public datasets using different prediction models. missForestPredict provides competitive results in prediction settings within short computation times.

## Introduction

Prediction models are based on various data sources, such as clinical data, customer-generated data, or data collected by electronic devices. Domains in which prediction models are used include disease diagnosis or prognosis, banking or financial applications such as fraud detection, and predictions based on sensor data. Often, datasets used to develop the models suffer from missing values: values not recorded or unavailable. Ignoring missing data by omitting rows in the dataset that have missing values (i.e. complete case analysis) or by omitting columns (predictor variables) with missing values can lead to limited applicability of the prediction model or to biased or suboptimal model performance. The model development phase

**Data availability statement:** All datasets used in current study are publicly available for research. Detailed information for each dataset is available in Supporting information. If not available directly in a R package, the datasets are also shared together with the code. The code used for running the comparisons is available at https://github.com/sibipx/comparison_imputation_methods. The code can be used as such by other researchers for comparing the imputation methods on their dataset, by following the instructions in the readme file. The code is organized in a modular fashion and experienced R users can add other prediction models on other imputation methods to compare.

**Funding:** This research was funded by KU Leuven funds (Grant number C24M/20/064 to LW and BVC). The funder had no role in the design of the study, the collection, analysis, and interpretation of data, or in writing the manuscript.

**Competing interests:** The authors have declared that no competing interests exist.

typically includes multiple data pre-processing steps, including missing data imputation, which refers to filling in missing values with plausible values based on observed information in the dataset.

When used in practice, prediction models are applied to make predictions for new observations. Many prediction applications are real time, predicting the outcome for a new observation as data become available. For example, a model implemented in electronic health record software may predict the occurrence of a health-related event for a patient as their data are recorded in the system. Some of the data needed to predict the event may not be available, for example due to a measurement device failing to record or transmit data, or due to unavailable patient history. Research on imputation methods has traditionally focused on the use of data for statistical inference, i.e. research on (causal) associations between measured variables and outcomes. Two main aspects differentiate applied prediction research from research that focuses on statistical inference: (1) the prediction model as well as any pre-processing steps (like missing data imputation) must be applicable to a single new observation and (2) no knowledge of the outcome is available at prediction time. Additional to these, other aspects that offer extra flexibility are desired: (3) support for both continuous and categorical variables and (4) allowing for missing data at prediction time in variables that were complete at development time.

The caret R package [1], the tidymodels R package [2] and the scikit-learn library in python [3] are generic libraries implementing tools supporting the complete pipeline for creating prediction models: data pre-processing (including missing data imputation), model building and model evaluation. As part of the pre-processing step, these frameworks support random forests [4] and kNN [5,6] based imputation methods. Alternatively, separate imputation libraries exist, such as the mice R package that implements MICE (multivariate imputation by chained equations) [7] and the missForest R package. missForest [8] is a popular imputation algorithm and its implementation in the R package is easy to use, though it does not support the imputation of new observations.

The current work introduces the extension of the missForest algorithm for prediction settings, supporting imputation of new observations at prediction time, available in missForestPredict R package. The performance of the missForestPredict algorithm in prediction settings is compared to the performance of other imputation methods that can impute a new observation at prediction time on eight simulated datasets with different missingness patterns, four public datasets with missing values and four complete datasets on which missing data are simulated. The use case of interest is that when data are missing both in the training set, as well as at prediction time, under the same missing data mechanism, and predictions on observations with missing values are of interest.

## Methods

### missForestPredict algorithm

missForestPredict extends on the algorithm implemented in missForest explained in [8]. The missing values of each variable in a dataset are first imputed with an initial

value. By default the mean/mode of each variable on complete cases of that variable is used as initialization, but median/mode or custom initialization schemes are also supported. Each variable is then imputed in an iterative manner using random forest (RF) models built on complete cases of that variable until a convergence criterion is met. At each iteration, these models are used to predict the values to be imputed for the missing values on each variable. The initial values and the random forest imputation models are saved for each variable and each iteration and can be later applied in the same order and for the same number of iterations for imputing new observations.

The algorithm steps are presented in Tables 1 and 2. $y^{(s)}$ represents the variable to be imputed and $x^{(s)}$ represents the other variables; s represents the sequence number of the variable. $y_{obs}^{(s)}$ represents the observed values on variable $y^{(s)}$. $x_{obs}^{(s)}$ represents the predictor matrix for $y_{obs}^{(s)}$ (all variables other than $y^{(s)}$ corresponding to observations that are observed on $y^{(s)}$). Similarly, $y_{miss}^{(s)}$ represents the missing observations on variable $y^{(s)}$, while $x_{miss}^{(s)}$ the predictor matrix for $y_{miss}^{(s)}$. $M_{is}$ represents the random forest model corresponding to variable with the sequence s at iteration i (Fig 1).

In random forests models, each tree is built on a bootstrap sample of the data, leaving out on average 36.8% of the data not used in learning that tree. Each observation will therefore not be used in around a third of the trees. Predictions for each observation are made using only the trees that did not use this specific observation for learning and are then averaged to obtain a final prediction, called out-of-bag (OOB) prediction. Using the OOB predictions, different metrics can be evaluated, generally named OOB error. The OOB error is considered a good estimation of the model error on unseen data [9,10]. The apparent error is the error calculated on each observation of the training set using the model learned on that training set.

missForestPredict uses by default the OOB error to determine the convergence of the algorithm. The apparent error and the out-of-bag (OOB) error for the observed part of the observations are saved for each variable and each iteration. At each iteration the normalized mean square error (NMSE) is calculated for each variable separately for both continuous and categorical variables. The NMSE can be interpreted as an improvement or deterioration of the imputations (with

**Table 1. Pseudocode for the `missForestPredict` algorithm.**

| missForestPredict algorithm |
| --- |
| INPUT: Dataframe with n rows and p columns; $\gamma$ <- convergence criterion value (based on NMSE) |
| Impute missing values with initialization scheme and save the initialization. |
| $k$ <- sorted columns in increasing amount of missingness |
| **while** not $\gamma$: |
| **for** $s$ in $k$: |
| Fit random forest model $y_{obs}^{(s)} \sim x_{obs}^{(s)}$ |
| save model $M_{is}$ |
| impute $y_{miss}^{(s)}$ using $x_{miss}^{(s)}$ |
| update $\gamma$ value |
| return $X^{imp}$ |
| return: initialization scheme; list of models; imputation sequence; no. of iterations $n$ |

**Table 2. Pseudocode for the `missForestPredict` algorithm - new observation.**

| missForestPredict - impute a new observation |
| --- |
| INPUT: one or more observations with missing values $X^{miss}$ |
| INPUT: missForestPredict object storing $M_{is}$ objects; imputation sequence $k$; initialization |
| Initialize observation based on initialization scheme. |
| for i from 1 to $n$: |
| for $s$ in $k$: |
| impute $y_{miss}^{(s)}$ using model $M_{is}$ |
| return imputed observation(s) $X^{imp}$ |

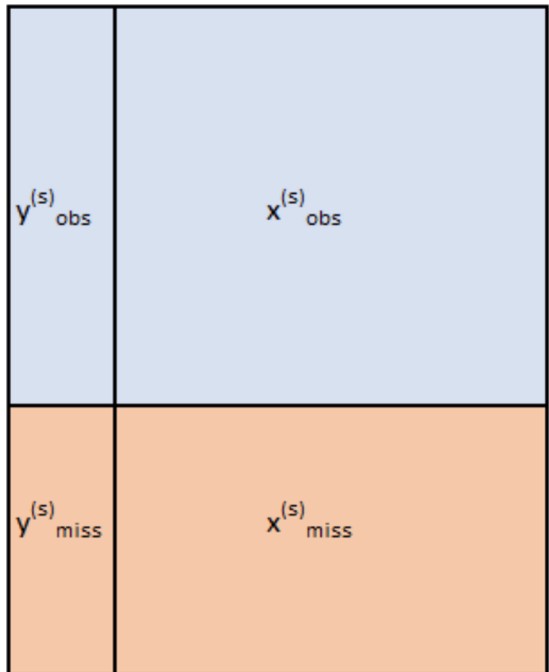

**Fig 1**. **Imputed variable and predictor matrix.**

regards to the distance from true values) compared to a reference imputation: the class proportion for categorical values and the mean imputation for continuous variables. Values smaller than 1 indicate an improvement compared to the reference imputation, while values higher than 1 indicate a deterioration.

**Continuous variables**:

For continuous variables, the NMSE is equivalent to $1-R^2$. Mean imputation has by definition an NMSE of 1.

$$NMSE = \frac{\sum_{i=1}^{N}(y_i-\hat{y}_i)^2}{\sum_{i=1}^{N}(y_i-\bar{y})^2} = 1 - R^2, \; i = 1, 2, ...N \; (1)$$

$y_i$ = the true value of variable y for observation i

$\bar{y}$ = the mean value of variable y

$\hat{y}_i$ = prediction (imputation) of variable y for observation i

$N$ = number of observations

**Categorical variables (including ordinal factors)**:

For categorical variables, NMSE is equivalent to $1-BSS$ (Brier Skill Score) or the Brier Score ($BS$) divided by the reference Brier Score ($BSref$). The Brier Score is calculated as the sum of square distances between the predictions (as probabilities) and the true values (0 or 1) for each class, and then summed up for all classes. The reference Brier Score is calculated as the Brier Score of a predictor that predicts the proportion of the event in each class [11].

$$NMSE = \frac{BS}{BSref} = 1 - BSS \; (2)$$

$$BS = \frac{1}{N}\sum_{j=1}^{R}\sum_{i=1}^{N}(p_{ij} - y_{ij})^2, \; i = 1, 2, ...N, j = 1, 2, ...R$$

$$BSref = \frac{1}{N}\sum_{j=1}^{R}\sum_{i=1}^{N}(p_j - y_{ij})^2 = 1 - \sum_{j=1}^{R}p_j^2$$

$y_{ij}$ = the true value of variable y for observation i and class j (1 if observation i has class j and 0 otherwise)

$p_{ij}$ = prediction (probability) for observation i and class j

$p_j$ = proportion of the event in class j

*N* = number of observations

*R* = number of classes

To obtain a global NMSE error a weighted average of the NMSE for all variables is calculated. By default the weight of each variable is set to the proportion of missing values for that variable, but the weights can be changed by the user. When the global NMSE in an iteration does not decrease compared to the global NMSE of the previous iteration, the algorithm stops and only models of previous iterations will be used in imputation. Users can inspect the variable-wise NMSE, as explained in the "missForestPredict convergence criteria and error monitoring" package vignette and visually assess the convergence. If visual inspection proves unsatisfactory, users can adapt either the weights of each variable in the convergence criterion (using var_weights parameter) or override the convergence logic by manually setting the number of iterations (using fixed_maxiter and maxiter parameters).

The following list summarizes the algorithmic aspects for missForestPredict:

- In one iteration, variables are imputed in a specific order (decreasing in proportion of missing by default). The imputation of each variable is done "on the fly". That is, on iteration i, the imputations of variable with sequence s will be available for the models built for variable with sequence s+1.
- The convergence criterion can be based on the OOB error (default) or on apparent error. The missForest package used the apparent error; we keep it as an option for users for backwards compatibility.
- The algorithm uses one unified convergence criterion for both continuous and categorical variables, based on NMSE.
- In order to calculate MSE/BS for categorical variables, internally probability forests are used (instead of classification forests) and the probabilities are then transformed to classes for returning the imputations.
- The imputation models can be saved for each variable and each iteration and can be applied later to new observations.

Additional to these highlights, the package includes a number of usability and functional improvements:

- missForestPredict uses the ranger package [12] to build the random forest models. Ranger provides a fast implementation of random forests, which comes with the advantage of substantially decreased computation time for the imputation.
- The default number of trees used in the RF imputation models is 500, the default of the ranger function (this can be changed by providing a different value for the num.trees parameter)
- The user can provide a custom initialization scheme based on domain knowledge or use the implemented mean/mode or median/mode initializations. (Examples are provided in the package vignettes)
- Additional metrics are provided for error monitoring: MSE and NMSE for continuous variables, MER (misclassification error rate) and F1 score for categorical variables with two categories and MER and macro F1 score for categorical variables with more than two categories. These are saved and returned for each iteration and can be plotted to assess the convergence of the algorithm. (Examples are provided in the package vignettes)
- The user can specify which variables to impute and which predictors to be used for the imputation of each variable by specifying a predictor matrix. By default imputation models are learned for all variables, regardless if they have missing values or not.
- The variables used in imputation can also be controlled by using the proportion_usable_cases parameter, which represents the variable-wise proportion of usable cases and contains two components: p_obs and p_miss. If *y* represents the variable to be imputed and *x* represents the variable used as predictor, p_obs represents the proportion of missing *x* among observed *y* and p_miss represents the proportion of observed *x* among missing *y*, as described in [13], with default values p_obs = 1 and p_miss = 0. If all values of a predictor are missing among the observed value of the variable to be imputed, the value of p_obs will be 1 and the model built will rely heavily on the initialized values. Similarly, if all values of a predictor are missing among the missing values of the outcome, p_miss will have a value of 0 and the imputations (predictions) will heavily rely on the initialized values.

## Comparison to alternative imputation methods

**Overview of methodology.** The primary goal is the comparison of the missForestPredict to other imputation methods based on random forests that offer implementations in R or python with applicability in prediction settings; more specifically, they can impute a new observation with missing data and they do not require knowledge of outcome (as does rfimpute, for example). We focus only on impute-then-predict methods implemented in published software libraries, and not on methods built in the prediction model (like random forests with surrogate splits), and we build four different prediction models (two regression models and two tree-based models) on the imputed data. The comparison is done in terms of variable-wise imputation performance (distance from true values on each variable) and prediction performance on a test set (the performance of a prediction model built on imputed train data and applied to imputed test data, where the test set is imputed using the imputation models learned on the train set). We compare the imputation methods on eight simulated datasets, on four different real complete datasets on which missing values are simulated and on four real datasets with missing values.

**Datasets.**

**Simulated datasets.** Four datasets with binary outcome and continuous predictor variables are simulated. The number of observations is 4000 in all datasets. Four continuous "signal variables" are simulated from a multivariate normal distribution with pair-wise correlation 0.1 (low but realistic correlation) or 0.7 (high correlation structure). Twelve additional "noise variables" are added with zero coefficient and zero correlation with the other variables. Enough noise variables are added to attempt to force RF based methods to do splits on noise variables and study the performance of these methods under noise conditions. The binary outcome is simulated using the logistic function with four equal coefficients corresponding to the signal variables and zero coefficients for the noise variables so that the area under the ROC curve (AUROC) is around either 0.75 (lower discrimination, realistic) or 0.9 (high discrimination scenario). The outcome prevalence is around 20%. The four coefficients for the predictor variables are optimized so that the desired discrimination is obtained using the method described in [14] and using the authors' shared code on github. From each dataset containing noise variables, a separate dataset is created by discarding the noise variables (Table 3). These datasets will be further studied under different missing data mechanisms (see section Amputation for details). The purpose of these datasets is to study the properties of the missForestPredict algorithm in comparison to other imputation methods in very simple controlled settings, even if these datasets are not tailored for random forest imputation methods (no complex relationships, like interactions and non-linearities, between variables).

**Real data.** The public datasets have been chosen from various repositories based on the following criteria: they should be large enough to produce sufficiently reliable results in prediction settings (as a rule of thumb we used datasets with more than 10000 observations; for datasets with binary outcome we ensured that at least 200 events will be present on average in the test set after a 2:1 train/test split) and they should have a clear outcome variable documented, either continuous or binary. Table 4 provides details on the complete datasets and the datasets with missing values. Max percent missing represents the missingness rate on the variable with most missing values. Because the diabetes dataset contains a large number of variables (27) out of which only few (9) have missing values, it is used twice: once as complete (the

**Table 3. Simulated datasets description.**

| Dataset | AUROC | Correlation between predictors | Noise |
|---|---|---|---|
| sim_75_1 | 0.75 | 0.1 | No |
| sim_75_7 | 0.75 | 0.7 | No |
| sim_90_1 | 0.90 | 0.1 | No |
| sim_90_7 | 0.90 | 0.7 | No |
| sim_75_1 noise | 0.75 | 0.1 | Yes |
| sim_75_7 noise | 0.75 | 0.0 | Yes |
| sim_90_1 noise | 0.90 | 0.1 | Yes |
| sim_90_7 noise | 0.90 | 0.7 | Yes |

**Table 4. Real datasets description.**

| Dataset | No. of obs. | No. of cont./categ. vars | Outcome (type, positive class proportion) | Source | Description | Max percent missing |
|---|---|---|---|---|---|---|
| **Complete** | | | | | | |
| diamonds | 53940 | 9/0 | price (continous) | ggplot2 R package | Attributes and prices of diamonds | |
| breast tumour | 58320 | 5/4 | target (continuous) | PMLB R package | Patient and breast tumour data | |
| Whitehall I | 17260 | 9/1 | 10-year mortality (binary, 9.7%) | Universitätsklinikum Freiburg | Medical and socioeconomical data on British Civil Servants | |
| diabetes | 49679 | 17/4 | 30-day readmission (binary, 11.3%) | UCI ML repo | Summaries of patient admissions for diabetic patients | |
| **With missing values** | | | | | | |
| covid | 15524 | 0/6 | result of SARS-CoV2 PCR test (binary, 7.5%) | medicaldata R package | Patient and test information for SARS-CoV2 testing | 45.65% |
| diabetes | 49679 | 19/8 | 30-day readmission (binary, 11.3%) | UCI ML repo | Summaries of patient admissions for diabetic patients | 94.95% |
| CRASH 2 | 20207 | 12/15 | 14-days in hospital mortality (binary, 15.2%) | Vanderbilt Biostatistics | Data on trauma patients | 49.31% |
| IST | 19410 | 5/20 | 14-days mortality (binary, 10.5%) | UK Medical Research Council, UK Stroke Association, European Union BIOMED-1 program | Medical data collected after acute ischaemic stroke event | 20.29% |

variables with missing values are removed) and once with the variables with missingness included. Detailed description of each dataset and data preparation steps (whenever applicable) are presented in S1 Appendix.

**Imputation methods.** Our primary goal is to compare missForestPredict to other imputation methods based on random forests that can impute a single new observation at prediction time. We focused on methods with implementation in R or python. A full overview of the methods considered for inclusion in the comparison and their evaluation is presented in S1 Appendix. Additional to these random forest imputation methods, we also compare with k-nearest neighbours (using the implementation in the tidymodels framework), linear regression imputation (tidymodels) and mean/mode imputation, and mice (using defaultMethod: "pmm" for continuous variables, "logreg" for categorical variables with two levels and "polyreg" for categorical variables with with more than two levels) because of their popularity and ready-to-use implementation. Details of the functions used and parameter settings for each method are presented in Table 5.

The **bagging** imputation method uses multiple trees fit on bootstrap samples to learn imputation models for each variable in the dataset using all the other predictors in the dataset. The *step_impute_bag* function uses the ipred package to learn the models. At prediction time, these models are used to impute a new observation. Compared to missForestPredict, bagging is not iterative (one single iteration is used), uses bagged trees (considering all variables for each split in a tree, as opposed to using only a fraction of variables as done in RFs) and does not require an initialization scheme.

**miceRanger** implements multiple imputation by chained equations (MICE) with random forests using the ranger R package while saving the models for imputation at a later stage. Initialization is done using random sampling from complete cases for each variable, after which random forest models are learned iteratively until a maximum number of iterations is reached. We use *maxiter* = 5, the default value in the package. The process is repeated *m* times, resulting in *m* imputed training sets. We use *m* = 5, the default value in the package. The imputed values of each observation on

**Table 5. Details on imputation methods in R or python.**

| Method | Package/library (version) | Function (train set) | Parameters | Function (test set) | R/python |
|---|---|---|---|---|---|
| **Tree based** | | | | | |
| bagging | tidymodels/recipes (1.0.8) | step_impute_bag | | bake | R |
| mice (RF) | mice (3.16.0) | futuremice | method = 'rf', ntree = 100, m = 5, maxit = 5 | mice.mids | R |
| miceRanger | miceRanger (1.5.0) | miceRanger | m = 5, maxiter = 5, num.trees = 100, parallel = TRUE, returnModels = TRUE | impute | R |
| IterativeImputer (shortname: python_II) | scikit-learn (1.4.1) | IterativeImputer. fit | n_estimators=100 | IterativeImputer. transform | python |
| missForestPredict (shortname: missFP) | missForestPredict (1.0) | missForest | num.trees = 100, initialization = 'mean/mode' | missForestPredict | R |
| missForestPredict_md10 (shortname: missFP_md10) | missForestPredict (1.0) | missForest | num.trees = 100, max.depth = 10, initialization = 'mean/mode' | missForestPredict | R |
| **Other** | | | | | |
| kNN | tidymodels/recipes (1.0.8) | step_impute_knn | neighbors = 5 | bake | R |
| linear | tidymodels/recipes (1.0.8) | step_impute_linear | | bake | R |
| mice (default) | mice (3.16.0) | futuremice | m = 5, maxit = 5 | mice.mids | R |
| mean/mode | custom implementation | impute_mean_ mode_train | | impute_mean_ mode_test | R |

each variable are aggregated across the 5 imputed datasets (using mean for continuous variables and mode for categorical variables) to obtain a single imputed dataset. The algorithmic differences between miceRanger and missForestPredict are: miceRanger implements natively multiple imputation (single imputation is obtained by averaging the imputed values); the initialization scheme is random sampling from complete cases (instead of mean/mode in missForestPredict); the maximum tree depth is fixed to 10 in miceRanger, while missForestPredict uses unlimited depth unless specified by the user in the max.depth parameter in ranger function; miceRanger uses by default predictive mean matching for the final imputed value; and miceRanger does not implement a convergence criterion and relies on a fixed number of iterations (5 by default).

**mice (method rf)** Multiple Imputation by Chained Equations (MICE) [15] is highly advocated in clinical inferential studies [16] as it results in unbiased regression coefficients and correct estimation of the coefficients' standard errors [17]. We use the implementation in the mice R package [18]. As in **miceRanger**, initialization is done using random sampling from complete cases for each variable, after which random forest models (using ranger by default) are learned iteratively until a maximum number of iterations is reached. The process is repeated $m$ times, resulting in $m$ imputed training sets. We use the default $maxit = 5$ of the $mice$ function for the number of iterations and the default $m = 5$ for the number of imputed datasets at training time. Although initially designed for imputing only one dataset of interest (on which inferences are made) without the possibility of applying imputations to a single new observation at prediction time, the $mice.mids$ function has been extended in the November 2020 CRAN release to impute observations that are not used for learning the imputation model. It uses a trained mids object (previously obtained using the $mice$ function) to which one or more new test observations are appended. Starting from the state of the already trained objects (trained with 5 iterations), it runs one more iteration of the mice algorithm (by default) for each of the $m = 5$ imputed datasets using only train data for learning the imputation; it then predicts on the appended test observation(s) using the learned models for this extra iteration. Although the number of additional iterations is configurable, we use the default $maxit = 1$ in the $mice.mids$ function. The differences between mice and missForestPredict are: mice implements natively multiple imputation (single imputation is

obtained by averaging the imputed values); the initialization scheme is random sampling from complete cases (instead of mean/mode in missForestPredict); mice uses predictive mean matching for the final imputed value; mice does not implement a convergence criterion and relies on a fixed number of iterations (5 by default); mice does not use the learned models from iterations 1 to 5 to impute new observations but it creates a sixth iteration for which the sixth imputation model is learned (starting from the fifth state of training) and it imputes the new observation using its initialization and the sixth imputation model.

The **IterativeImputer** in the sci-kit learn library in python implements the same iterative algorithm as missForestPredict using by default mean initialization. Unlike missForestPredict and miceRanger, it only supports continuous variables. Similar to missForestPredict, it uses a convergence criterion to control the number of iterations; the convergence criterion is based on the decrease in apparent error. The algorithmic differences to missForestPredict are: IterativeImputer does not support categorical variables; the convergence criterion is based on absolute differences from previous imputation (within a tolerance limit with default value of 0.001); the maximum number of iterations is by default 10.

**missForestPredict** imputation is run twice: once with default settings (missForestPredict with deep trees) and once setting max.depth parameter of ranger function (maximum depth of the trees) to 10 (missForestPredict with shallow trees). Constraining the maximum depth has the advantage of improved computation times and reduction in memory usage for the stored imputation models; it is also the value used in miceRanger and allows us to compare the two algorithms in similar settings.

**k nearest neighbours** imputes missing data by finding the k nearest neighbours from the training set and taking the mean value on each variable. For each observation to be imputed, the k nearest neighbours are selected from the full set of "donors" (given an observation with a missing value in one variable and observed values for a set of other variables, the set of donors consists of observations that also have observed values on the same set of variables, as well as on the variable to be imputed). The *step_impute_knn* function uses the Gower's distance [19] to find the nearest neighbours, supporting both continuous and categorical variables. The method does not involve an imputation model "fit", as neighbours can be found "on the fly" in the training set for each new observation. The default value of $k = 5$ neighbours is used. kNN does not learn an imputation model and requires the full training set at prediction time for matching neighbours for new observations.

**Linear regression imputation** (referred to as **linear imputation** hereafter) is done using the implementation in tidymodels, which supports only continuous variables and uses the *lm* function (assuming linear associations) to fit the imputation models. Categorical variables are therefore binarized and imputed as continuous (see below on binarization). By default, using variables with missing values as predictors for a variable to be imputed is not supported. To work around this limitation, we first initialize the missing values with mean/mode imputation; then, for each variable to be imputed we retain separate dataframes formed of this variable with missing values and the other variables initialized with mean/mode and use the *step_impute_linear* function for imputation. We then merge the separately imputed variables in a complete dataframe. The test set variables are initialized using the mean/mode from the train set and imputed using the linear models learned for each variable.

**mice (default method)** The implementation is similar to **mice (method rf)** but instead of using RF models, we use the default methods of the *mice* function: "pmm" (predictive mean matching) for continuous variables, "logreg" (logistic regression) for categorical variables with with two levels and "polyreg" (polytomous logistic regression) for categorical variables with more than two levels.

Finally, for **mean/mode imputation**, the mean value of complete cases for continuous variables or the most frequent value of complete cases for categorical variables are used to impute the missing values. The mean/mode for each variable on the training set are saved and used later on the test set.

On some datasets we impute variables with low missingness rate and on some random train/test splits these variables are complete in the training set but missing in the test set. Most imputation methods allow the user to specify the list of variables to impute, regardless of whether the variables have missing values or not, and we make use of this functionality.

The mice implementation will only impute at training time the variables with missing values. At test time though it deals gracefully with this situation: if the new observation(s) appended have missing values in variables not missing at training time, models will be learned for these variables as well during the additional iteration. Iterative Imputer only implements a skip_complete parameter, which will skip learning imputation models for the complete variables in the training set. This will result in failure of imputing the test set, therefore we do not use the skip_complete parameter. In consequence, Iterative Imputer will be the only method building imputation models for all variables in a dataset.

The default number of trees used by the RF imputation methods are: 10 for mice, 25 for bagging, 100 for IterativeImputer (using RandomForestRegressor) and 500 for miceRanger and missForestPredict. We have set the number of trees to 100 for all RF based imputation methods to optimize the memory usage for the methods using 500 trees and to bring on par the methods that use less trees. All other parameter settings have been left on default values for all methods, including missForestPredict. We have though added missForestPredict with max depth 10 to study its impact on runtime, memory utilization and imputation performance on large datasets.

**Amputation methods.** The process of creating missing values, also called "amputation", can follow different missing data mechanisms, namely missing completely at random (MCAR), missing at random (MAR), and missing not at random (MNAR), as introduced in [17]. These missing data mechanisms are widely studied in inferential statistics. MAR missing data mechanism in which the missingness depends on the outcome values and MNAR mechanism are expected to create bias in the coefficients of regression models [20]. In the context of regression prediction models, this bias in the coefficients estimates could increase the prediction error (in addition to the irreducible error and the imputation error) affecting the overall predictive performance of the model. We will study the impact of missing data mechanisms only on the simulated datasets. For the real datasets without missing values, missingness is created only following the MCAR mechanism, as creating realistic MAR or MNAR mechanisms involves in depth knowledge of each dataset.

For the simulated datasets, the amputation mechanisms are described in Table 6. MAR and MNAR missing data mechanisms are simulated only for "signal" variables. Noise variables are always "amputed" using MCAR missingness, regardless of the missing data mechanism used for the signal variables. This is done to avoid that the missingness mechanism adds "signal" to these variables either for the imputation model or for the prediction model. The goal is to keep these variables as "pure noise" at all levels of the workflow.

**Variable transformations.** missForestPredict, mice, miceRanger, bagging, kNN (tidymodels) and mean/mode imputation can impute both continuous and categorical variables. IterativeImputer and linear imputation (tidymodels) can only impute continuous variables. To put the methods on par, the categorical variables (with two or more categories) are binarized (dummy-coded). This transformation is applied after the values on the original categorical variable have been "amputed". (details in S1 Appendix) For simplicity, ordinal variables will be transformed to continuous variables and used as continuous both in the imputation models as well as in the prediction model.

**Prediction models.** Four prediction models to predict the outcome (binary or continuous) are built on each imputed training set: a random forest (RF) model, a gradient boosting (XBG) model using xgboost [21], an L2-regularized (logistic) regression model (Ridge) and a restricted cubic spline (logistic) regression model (Cubic spline). All models except the restricted cubic spline model need hyperparameter tuning. The tuning is done such that it optimizes the logloss for binary outcomes and RMSE for continuous outcomes. Details on tuning and model building can be found in S1 Appendix.

**Comparison procedure.** The prediction performance comparison is performed on a number of datasets with missing values or on complete datasets (real and simulated) with simulated missingness. The steps performed are in a large part the same, with the exception that for datasets with missing values no 'amputation' is necessary and the variable-wise performance cannot be evaluated, as the true values are unknown. On each dataset and for each imputation method, the steps detailed in Table 7 are preformed 100 times.

The runtimes for fitting the imputation models, for imputing the train sets and for imputing the test sets, and the stored object sizes for the R imputation models are also saved and compared. The comparisons have been run on a high-performance computing cluster using 36 cores and 128 GB RAM using R version 4.2.1 and python 3.10.8.

**Table 6. Amputation mechanisms.**

| Name | Description |
|---|---|
| MCAR | A random sample of size corresponding to 30% of the observations is set to missing on each variable. |
| MAR_2 | Variable V1 is amputed in function of values of variable V2: A random sample of size corresponding to 10% of the observations for which values of V2 are lower than the mean of V2 is set to missing. A random sample of size corresponding to 50% of the observations for which values of V2 are greater than the mean of V2 is set to missing. In the same manner, variable V3 is amputed in function of values of variable V4. This procedure results in around 30% missingness on two variables, while leaving the other two variables complete. |
| MAR_2_out | Variable V1 is amputed in function of values of variable V2 and the outcome variable: A random sample of size corresponding to 10% of the observations for which values of V2 are lower than the mean of V2 and outcome is in positive class is set to missing . A random sample of size corresponding to 36% of the observations for which values of V2 are lower than the mean of V2 and outcome is in negative class is set to missing. A random sample of size corresponding to 20% of the observations for which values of V2 greater are than the mean of V2 and outcome is in positive class is set to missing. A random sample of size corresponding to 30% of the observations for which values of V2 greater are than the mean of V2 and outcome is in negative class is set to missing. In the same manner, variable V3 is amputed in function of values of variable V4. This procedure results in around 30% missingness on two variables, while leaving the other two variables complete. |
| MAR_circ | Missingness is created in all four signal variables in the same manner for MAR_2 amputation mechanism, amputing variable V1 in function of V2, V2 in function of V3, V3 in function of V4 and V4 in function of V1. This procedure results in around 30% missingness on all variables. |
| MAR_circ_out | Missingness is created in all four signal variables in the same manner for MAR_circ amputation mechanism, but creating missingness on each variable in function of the next variable and the outcome (as in scenario MAR_2_out). This procedure results in around 30% missingness on all variables. |
| MNAR | Missingness is created in all four signal variables in with probability 10% in the lower than mean segment of variable values and 50% in the greater than mean segment of variable values. |

**Considerations for failure situations.** When creating random missingness on each variable independently, it is possible that observations with missing values on all variables in the dataset are created. This is more likely for datasets with smaller numbers of variables. kNN algorithm cannot deal with such a "completely missing" situation, as no match can be done on variables with non-missing values. These observations are deleted from the train or test set in which they occur, to allow kNN to succeed in imputing the dataset. Imputation methods that use an initialization scheme do not have these drawbacks, even if imputation of such a "completely missing" observation will be far from optimal.

A summary of the average number of completely missing observations over all train/test splits for each dataset is presented in Table 8. Datasets with less variables have higher chance of resulting in completely missing observations and consequently the simulated datasets with four variables (without noise) will be subject to such deletions, while the datasets with noise variables (16 variables) do not encounter this situation. Even if the first four variables are identical in the datasets with or without noise variables, the train/test splits are identical and the missing data simulation procedure will create missingness on the same observations, small variations in results on datasets with vs. without noise will occur due to this deletion. The proportion of complete cases (after the previously mentioned deletions) per dataset are presented in the S1 Appendix file.

## Results

The results presented here as well as additional evaluation metrics (AUROC, AUPRC, calibration measures, RMSE, MAE, MAPE, SMAPE for prediction performance) are available at: https://sibip.shinyapps.io/Results_imputation_methods/.

### Results on simulated datasets with simulated missingness (amputation)

**Datasets with low correlation (0.1) and low AUROC (0.75).** In low correlation settings, none of the imputation methods noticeably outperforms mean/mode imputation in terms of variable-wise deviations form true values or predictive performance. Mean/mode, linear and bagging produce the best variable-wise results. missForestPredict over-fits imputations ("over-imputes") in some but not all scenarios, while mice (default and RF), kNN, Iterative Imputer (python_II)

**Table 7. Comparison procedure steps.**

| Step | Datasets with simulated missingness | Datasets with missing values | Description |
|------|-------------------------------------|------------------------------|-------------|
| Train/test split | YES | YES | The dataset is randomly split in training set and test set, where the test set contains a third of the observations. |
| 'Ampute' both the training and the test set | YES | NO | For real datasets, 30% missing values are randomly created on each predictor variable independently, assuming MCAR. For simulated datasets, MAR, MCAR and MNAR missingness is simulated as explained in the Amputation section, so that around 30% missingness is created on each variables (except MAR_2 scenarios, for which only two variables have 30% missingness). The outcome variable is not 'amputed'. |
| Apply variable transformation | YES | YES | The categorical variables are transformed to binary variables as described above (if necessary for the imputation method) |
| Impute train set and learn imputation models for all variables | YES | YES | The predictor variables are imputed using each method on the training set (the outcome variable is not included in the imputation model). For miceRanger we retain the final OOB errors; for missForestPredict the final OOB errors and the number of iterations until convergence are retained. |
| Train prediction model on the imputed dataset | YES | YES | On the imputed training set, four prediction models are fit (RF, XGB, Ridge, Cubic spline). |
| Train prediction model on the original dataset | YES | NO | The prediction models are also learned on the original training set (with no 'amputed' values) |
| Impute test set | YES | YES | The imputation models learned on the training set are applied to the observations in the test set |
| Evaluate predictions on the imputed test set | YES | YES | Using the prediction models trained on the train set make predictions for each observation in the test set and evaluate the predictive performance using various performance metrics. In the main text we present the R-squared for continuous outcomes and the Brier Skill Score (BSS) for binary outcomes. Additional metrics, like the area under the ROC curve (AUROC) and the calibration metrics are presented in an external Shiny app. |
| Evaluate predictions on the original test set | YES | NO | Predictions will also be made and evaluated on the original complete test set (with no 'amputed' values) using the models learned on the original train set. |
| Evaluate imputation error on each variable | YES | NO | On the test set, evaluate the distance between the true value and the imputed value for each variable using NMSE for continuous variables and MER (misclassification error rate) for categorical variables. |

**Table 8. Mean number of completely missing observations over all train/test splits for datasets with simulated missingness (amputation).**

| Dataset | No. (proportion) completely missing obs. - train | No. (proportion) completely missing obs. - test |
|---|---|---|
| breast tumor | 0.77 (0) | 0.44 (0) |
| diamonds | 0.72 (0) | 0.27 (0) |
| whitehall1 | 0.06 (0) | 0.05 (0) |
| sim_75_1 - MAR_circ | 25.2 (0.01) | 13.66 (0.01) |
| sim_75_1 - MAR_circ_out | 26.9 (0.01) | 13.16 (0.01) |
| sim_75_1 - MCAR | 20.77 (0.01) | 10.53 (0.01) |
| sim_75_1 - MNAR | 24.33 (0.01) | 12.15 (0.01) |
| sim_75_7 - MAR_circ | 51.08 (0.02) | 24.87 (0.02) |
| sim_75_7 - MAR_circ_out | 27.07 (0.01) | 14.16 (0.01) |
| sim_75_7 - MCAR | 20.77 (0.01) | 10.53 (0.01) |
| sim_75_7 - MNAR | 51.84 (0.02) | 25.05 (0.02) |
| sim_90_1 - MAR_circ | 24.83 (0.01) | 12.88 (0.01) |
| sim_90_1 - MAR_circ_out | 26.55 (0.01) | 13.43 (0.01) |
| sim_90_1 - MCAR | 20.77 (0.01) | 10.53 (0.01) |
| sim_90_1 - MNAR | 25.41 (0.01) | 12.48 (0.01) |
| sim_90_7 - MAR_circ | 54.57 (0.02) | 26.07 (0.02) |
| sim_90_7 - MAR_circ_out | 27.78 (0.01) | 14.5 (0.01) |
| sim_90_7 - MCAR | 20.77 (0.01) | 10.53 (0.01) |
| sim_90_7 - MNAR | 52.78 (0.02) | 26.56 (0.02) |

and miceRanger "over-impute" in all scenarios. Adding noise seems slightly beneficial for the methods that over-impute in terms of variable-wise NMSE, but without visible impact on the prediction performance. The methods that over-impute produce lower prediction performance in all scenarios.

Mean/mode, linear and bagging imputation methods produce variable-wise NMSE close to one on all four variables in all missing data scenarios, except MNAR (Fig 2); this indicates that these methods impute values that are close to the mean. In the MNAR scenario, the mean of the train set deviates from the mean of the test set due to the amputation procedure, but these methods produce test NMSE lower than the other methods. missForestPredict with deep trees (missFP) also imputes values closer to the mean in all scenarios except MAR_circ (the most complex scenario), for which values deviate slightly from the mean (NMSE greater than one). missForestPredict with shallow trees (missFP_md10) deviates slightly from the mean of the train set in all scenarios. This behaviour is explained by the fact that missForestPredict with deep trees converges at iteration one, while missForestPredict with shallow trees runs for a number of iterations before convergence (S1 Appendix). mice (default and RF), kNN, Iterative Imputer (python_II) and miceRanger produce NMSE greater than one (imputations worse than mean imputation). Adding noise has negligible effect for the imputation methods that produce an NMSE close to one (mean/mode, linear and bagging), but affects to some extent kNN, Iterative Imputer, miceRanger and missForestPredict, for which noise seems slightly beneficial (lower NMSE, closer to one). This effect is most visible in the MAR_circ scenario. missForestPredict tends to converge faster in the presence of noise variables (S1 Appendix), suggesting that adding noise has a regularizing effect preventing the algorithm from over-imputing.

For the outcome independent amputation scenarios (MCAR, MAR_2, MAR_circ and MNAR), mean/mode, linear, bagging and missForestPredict with deep trees generally have better prediction performance than the other methods (Fig 3). The largest difference in BSS within model and amputation method, of 0.017 is encountered for the XGB model with mice (RF) having a BSS of 0.082 and mean/mode a BSS of 0.099.

For the outcome dependent amputation scenarios (MAR_2_out and MAR_circ_out), mean/mode, linear, bagging and missForestPredict with deep trees perform best and exceed the BSS of the original dataset for all prediction models. The differences are larger for tree based prediction models (RF and XGB) than linear prediction models (Ridge and cubic splines), e.g.: for XGB and MAR_circ_out, mean/mode has a BSS of 0.176 and missForestPredict 0.178 compared to

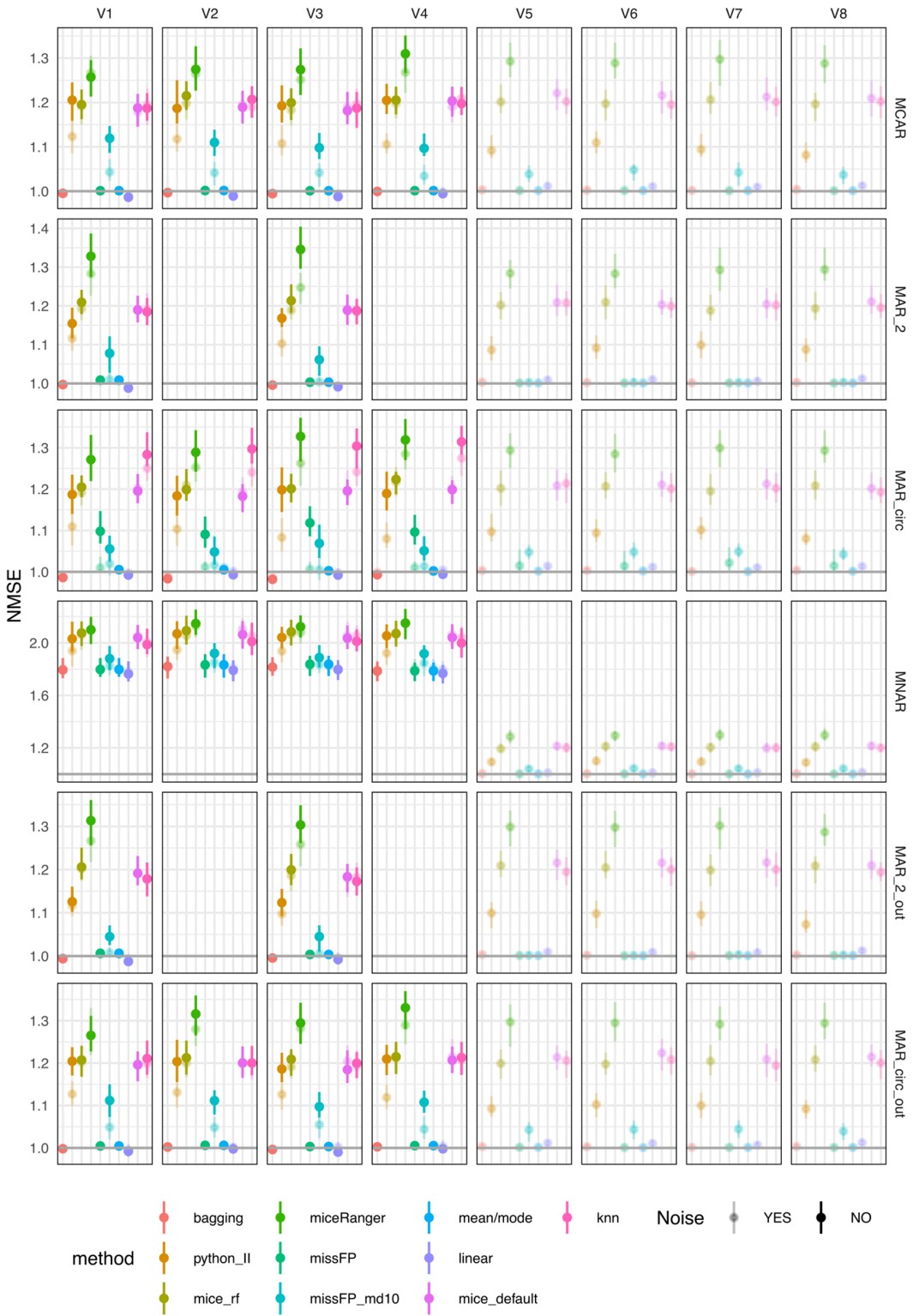

**Fig 2. NMSE errors (deviations from true values) on test sets for simulated datasets with simulated missingness: low correlation (0.1) and low AUROC (0.75).** To facilitate visualisation, only four out of the twelve noise variables are included in the figure.

**Fig 3. Prediction performance (BSS) for simulated datasets with simulated missingness: low correlation (0.1) and low AUROC (0.75).**

0.1384 on the original dataset; for the cubic splines model and MAR_circ_out, mean/mode has a BSS of 0.1526 and miss-ForestPredict 0.1526 compared to 0.1474 on the original dataset.

In all missing data scenarios and for all imputation methods, the prediction performance on the datasets with noise variables is slightly lower than the performance on datasets without noise, with cubic splines and RF models showing a bigger difference and Ridge and XGB being more resistant to noise; the impact of noise might be due to model's sensitivity to noise, rather than due to the imputation. The prediction performance results correlate with the results on the deviation from the true values for all scenarios, with mean/mode, linear, bagging and missForestPredict with deep trees imputation methods having both the lowest variable-wise NMSE and highest BSS.

The OOB NMSE for missForestPredict with shallow trees displays more bias than for missForestPredict with deep trees (S1 Appendix), with the largest bias for the MNAR scenario. miceRanger OOB bias is larger than the two missForestPredict imputation strategies. missForestPredict with deep trees converges after the first iteration in all scenarios except MAR_circ (S1 Appendix), indicating that the final imputation is essentially the initialization, which is mean/mode. Despite some differences in OOB bias, iterations until convergence and variable-wise NMSE, the differences in prediction performance between missForestPredict with deep trees and missForestPredict with shallow trees are generally small except for tree based prediction models (RF, XGB) and outcome related missingness, where growing deep trees displays an advantage, as the algorithm converges after one iteration and the final imputation is the mean/mode imputation.

**Datasets with low correlation (0.1) and high AUROC (0.9).** As the correlation between the four variables in the datasets is the same as in the previous scenarios (0.1), and the outcome is not included in the imputation, the variable-wise NMSE results (Fig 4) are very similar to the previous results (low correlation and low AUROC).

The prediction performance results (Fig 5) are also similar, with mean/mode, linear, bagging and missForestPredict performing better than the other methods, but the BSS differences between imputation methods are larger, e.g.: for MNAR the largest difference of 0.057 is encountered for the XGB model with mice (RF) having a BSS of 0.216 and missForestPredict with deep trees imputation a BSS of 0.273.

The prediction performance results also correlate with the variable-wise NMSE. As before, cubic splines and RF models are impacted more than Ridge and XGB models when adding noise variables. For the outcome related missingness, the prediction performance on imputed datasets does not though exceed the performance on the original dataset.

The differences between missForestPredict with deep and shallow trees are more pronounced, especially in the MAR_circ_out scenario, in the advantage of missForestPredict with deep trees, which mostly converge on first iteration (S1 Appendix).

**Datasets with high correlation (0.7) and low AUROC (0.75).** In high correlation settings, linear imputation and missForestPredict produce the imputations closest to the true values. The presence of noise variables impacts the variable-wise NMSE and prediction performance in a detrimental way, with kNN being the most impacted. In the outcome independent missingness scenarios, mean/mode and mice (default and RF) generally produce lower predictive performance, while the differences between the other imputation methods are small. In the outcome dependent scenarios, mean/mode performs best.

The variable-wise test NMSE for all methods is less than one in all scenarios, except mean/mode imputation which is higher than one in the MNAR, MAR_2 and MAR_circ scenarios and close to one in the MCAR, MAR_2_out and MAR_circ_out scenarios and mice (default and RF) for which NMSE is close to one in the MNAR scenario (Fig 6). For MAR_2 and MAR_circ the mean learned on the training set is biased. This is an artifact of the simulated MAR scenarios: the missingness rate is 50% in the higher than mean values of the variable and 10% in the lower than mean values. On the contrary, for the MAR scenarios outcome related, the missingness in higher and lower values is roughly equal. The presence of noise variables impacts the variable-wise NMSE in a detrimental way in the scenarios when missingness is present in all four variables (MCAR, MAR_circ, MNAR and MAR_circ_out) and these are imputed based on variables with missing values; the two scenarios with missingness on only two variables (MAR_2 and MAR_2_out) are not heavily impacted by noise. Across all imputation methods and all missing data scenarios, kNN is most impacted by the added

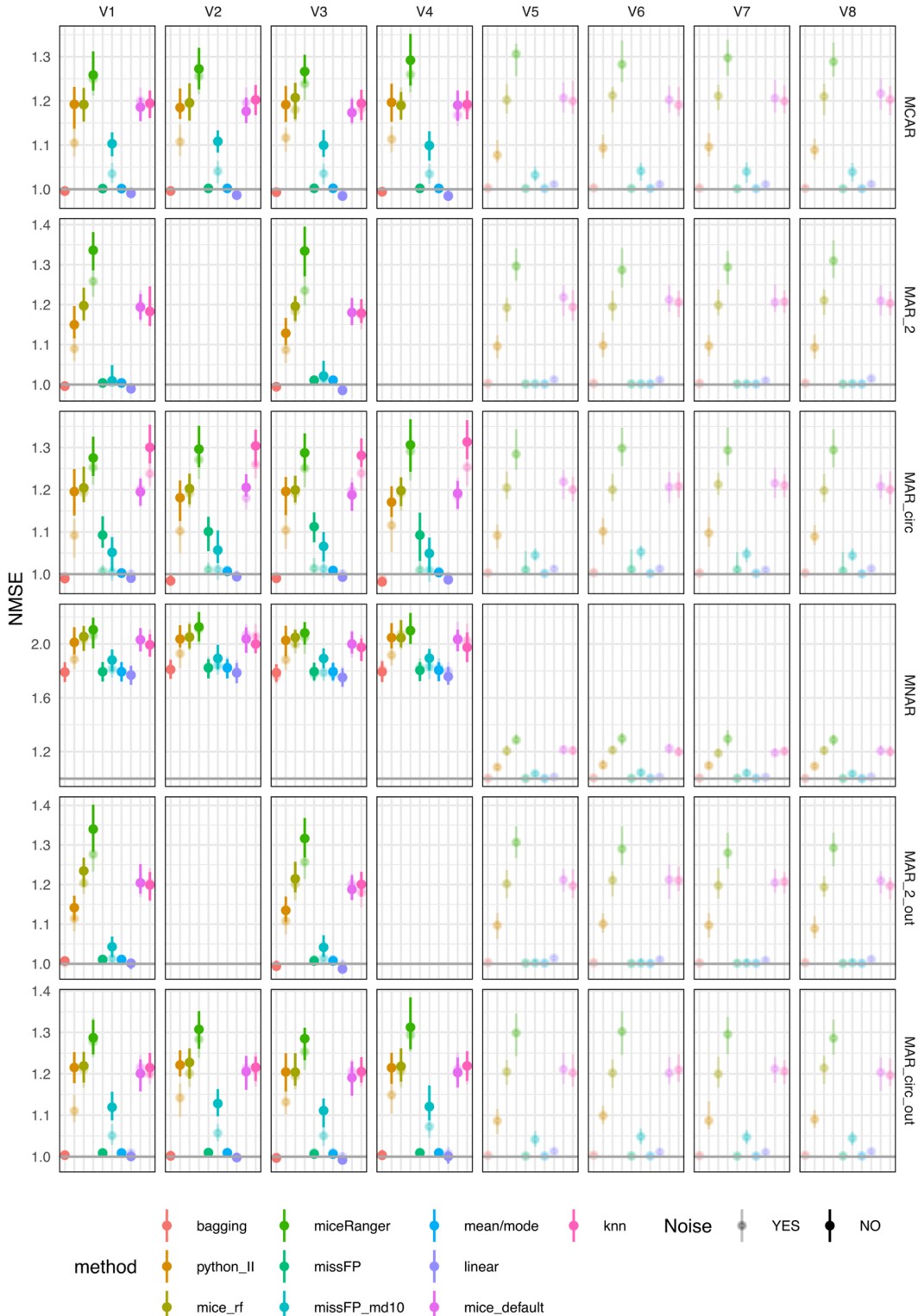

**Fig 4**. **NMSE errors (deviations from true values) on test sets for simulated datasets with simulated missingness: low correlation (0.1) and high AUROC (0.9).** To facilitate visualisation, only four out of the twelve noise variables are included in the figure.

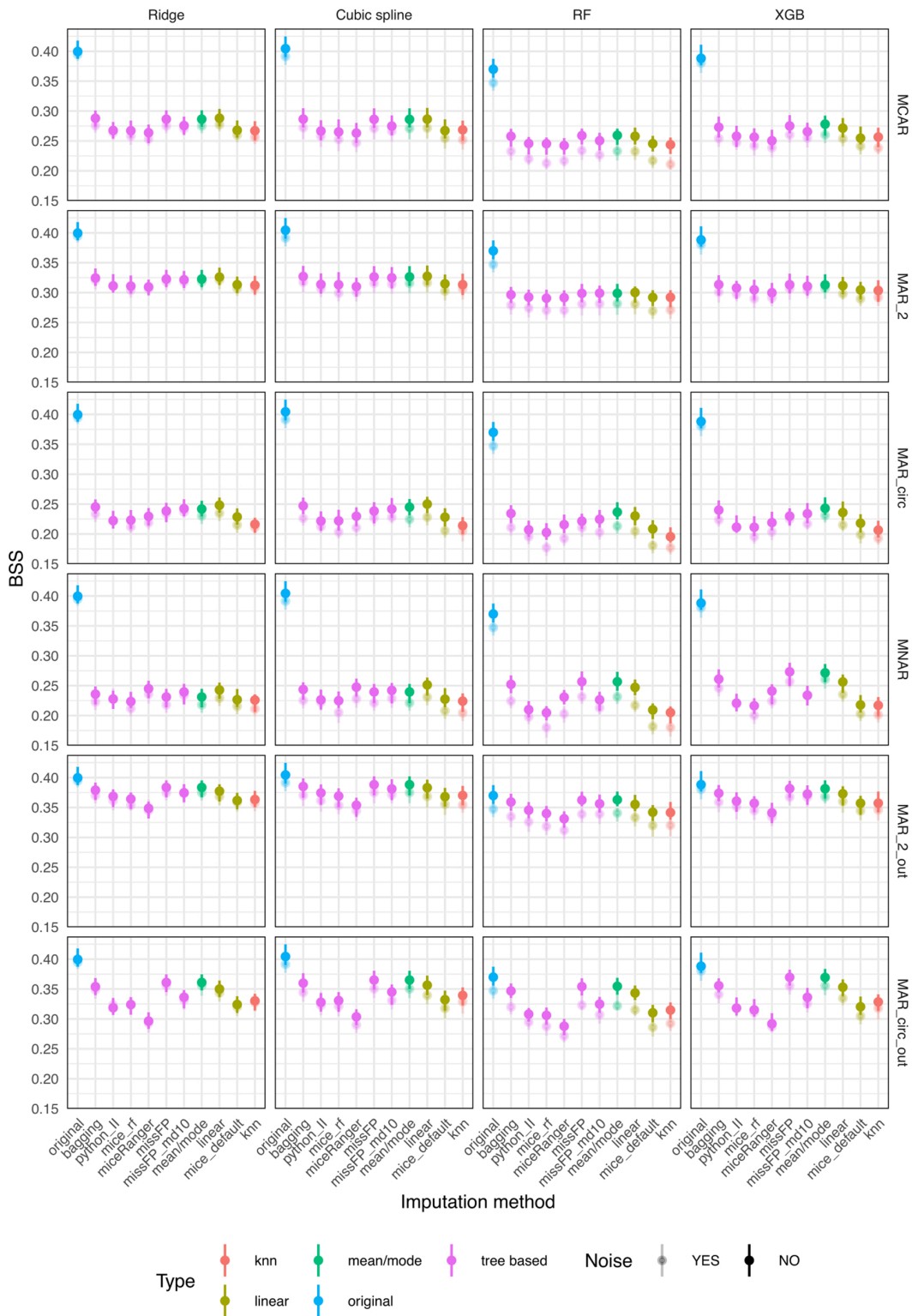

**Fig 5. Prediction performance (BSS) for simulated datasets with simulated missingness: low correlation (0.1) and high AUROC (0.9).**

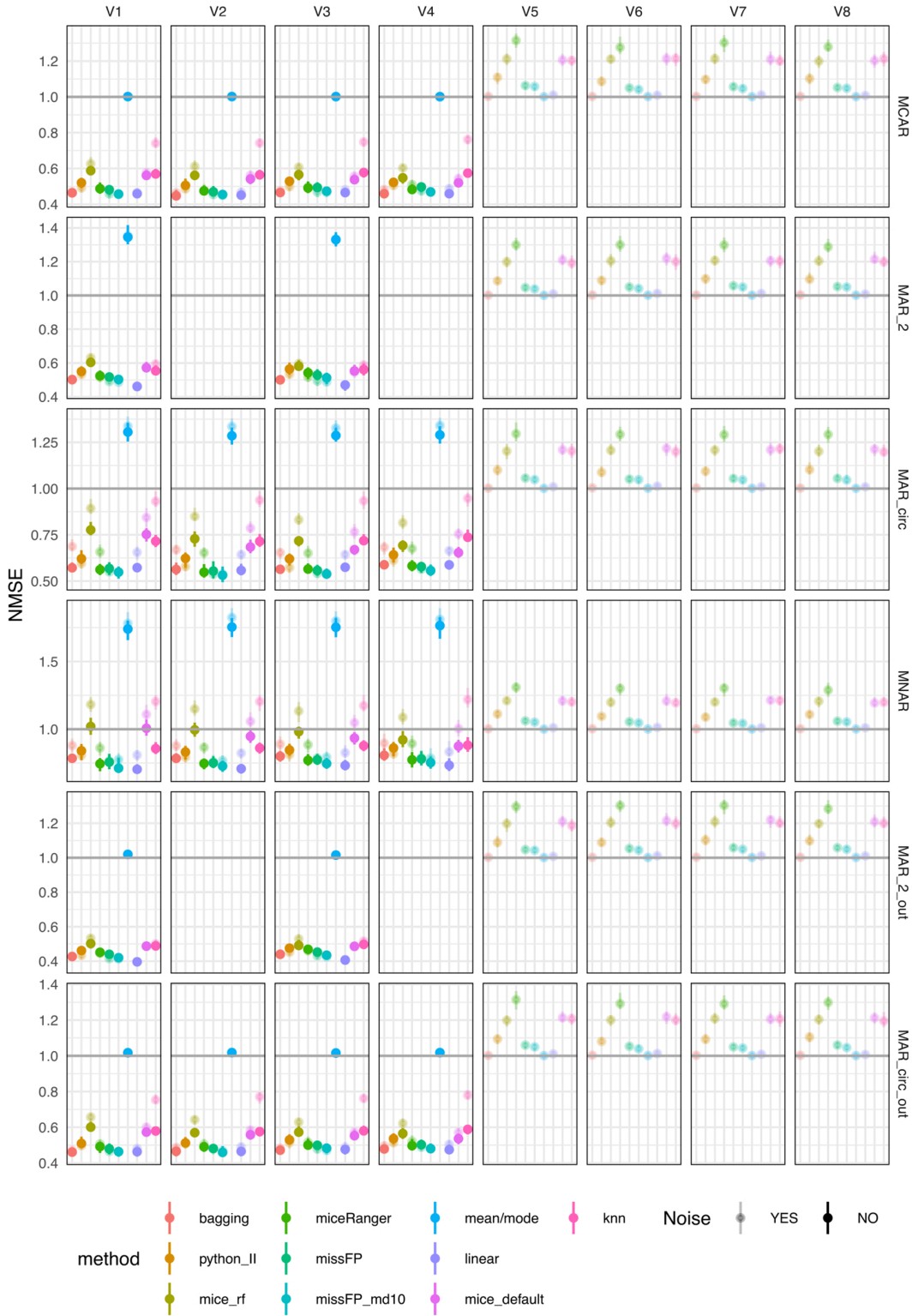

**Fig 6**. **NMSE errors (deviations from true values) on test sets for simulated datasets with simulated missingness: high correlation (0.7) and low AUROC (0.75).** To facilitate visualisation, only four out of the twelve noise variables are included in the figure.

noise variables, and mean/mode, missForestPredict and Iterative Imputer the least impacted. To be noted that the small variations in the mean/mode results are an artifact of the aforementioned deletions (Table 8). Across all scenarios, linear imputation and missForestPredict with shallow trees produce the lowest NMSE. Bagging produces low NMSE, similar to linear imputation and missForestPredict on all scenarios except MNAR, on which miceRanger produces low NMSE.

In outcome independent missingness scenarios (MCAR, MAR_2, MAR_circ and MNAR), mean/mode imputation yields inferior results in terms of predictive performance, with the difference being more pronounced for the regression-based prediction models: Ridge and cubic splines (Fig 7). The differences between imputation methods are more pronounced in the MAR_circ and MNAR scenarios. The largest difference in BSS within model and amputation method, of 0.021 is encountered for the Cubic spline model with mean/mode having a BSS of 0.117 and missForestPredict with shallow trees a BSS of 0.137.

For outcome related missingness scenarios, the mean/mode imputation performs the best and exceeds the performance of the original dataset, with differences more pronounced in the tree-based prediction models (RF and XGB), e.g.: for MAR_circ_out and Ridge model mean/mode produces a BSS of 0.1631 compared to 0.1527 on original dataset, and for the RF model mean/mode produces a BSS of 0.1882 compared to 0.1483 on original dataset. In the MAR_circ_out scenario, linear and bagging imputation methods (which are simpler, non-iterative imputation methods) provide a slight advantage compared to the other imputation methods, especially for tree based prediction models (RF and XGB).

The impact of noise variables is more pronounced than in the low correlation datasets and is more noticeable in the more drastic amputation scenarios (MAR_circ and MNAR). Among imputation methods, kNN seems the most impacted by noise for all prediction models. The prediction performance results correlate to some extent to the variable-wise imputation performance. For example, for MAR_circ and MNAR, mean/mode, mice (with default parameters) and mice (RF) produce the highest NMSE and the lowest BSS.

There is no notable difference between missForestPredict with deep and shallow in prediction performance or distance from true values. The OOB bias for the two missForestPredict imputation methods (S1 Appendix) is similar and comparable to the OOB bias of miceRanger. The differences in iterations until convergence are also small, both converging around 4-5 iterations. (S1 Appendix).

**Datasets with high correlation (0.7) and high AUROC (0.9).** As the correlation between the four variables in the datasets is the same as in the previous scenario (0.7), the variable-wise NMSE results (Fig 8) are very similar to the previous results (low correlation and low AUROC).

The prediction performance results (Fig 9) are also similar, with bagging, miceRanger and missForestPredict performing best and mice (default and RF) producing inferior results in the outcome independent missingness scenarios. Mean/mode also provides inferior prediction performance for linear models, but for tree based models (RF, XGB) it only produces inferior results in the simpler missingness scenarios (MCAR, MAR_2) but not in the more complex missingness scenarios (MAR_circ, MNAR). In these more complex missingness scenarios, bagging, miceRanger and missForestPredict produce the best results.

In the outcome dependent missingness scenarios, mean/mode provides the best results, close to or exceeding the performance on the original dataset, followed by linear and bagging imputation.

The impact of noise variables is more pronounced in the MAR_circ_out and MNAR scenarios and is highest for kNN and lowest for Iterative Imputer and missForestPredict.

As in the previous scenario, there is no notable difference between missForestPredict with deep vs. shallow trees in prediction performance, variable-wise deviation from true values, OOB bias or iterations until convergence (S1 Appendix).

## Results on real datasets with simulated missingness (amputation)

The variable-wise deviation from true values are presented as NMSE for continuous variables and misclassification error rare (MER) for categorical variables (Fig 10). Linear, bagging and missForestPredict yield best variable-wise results. The

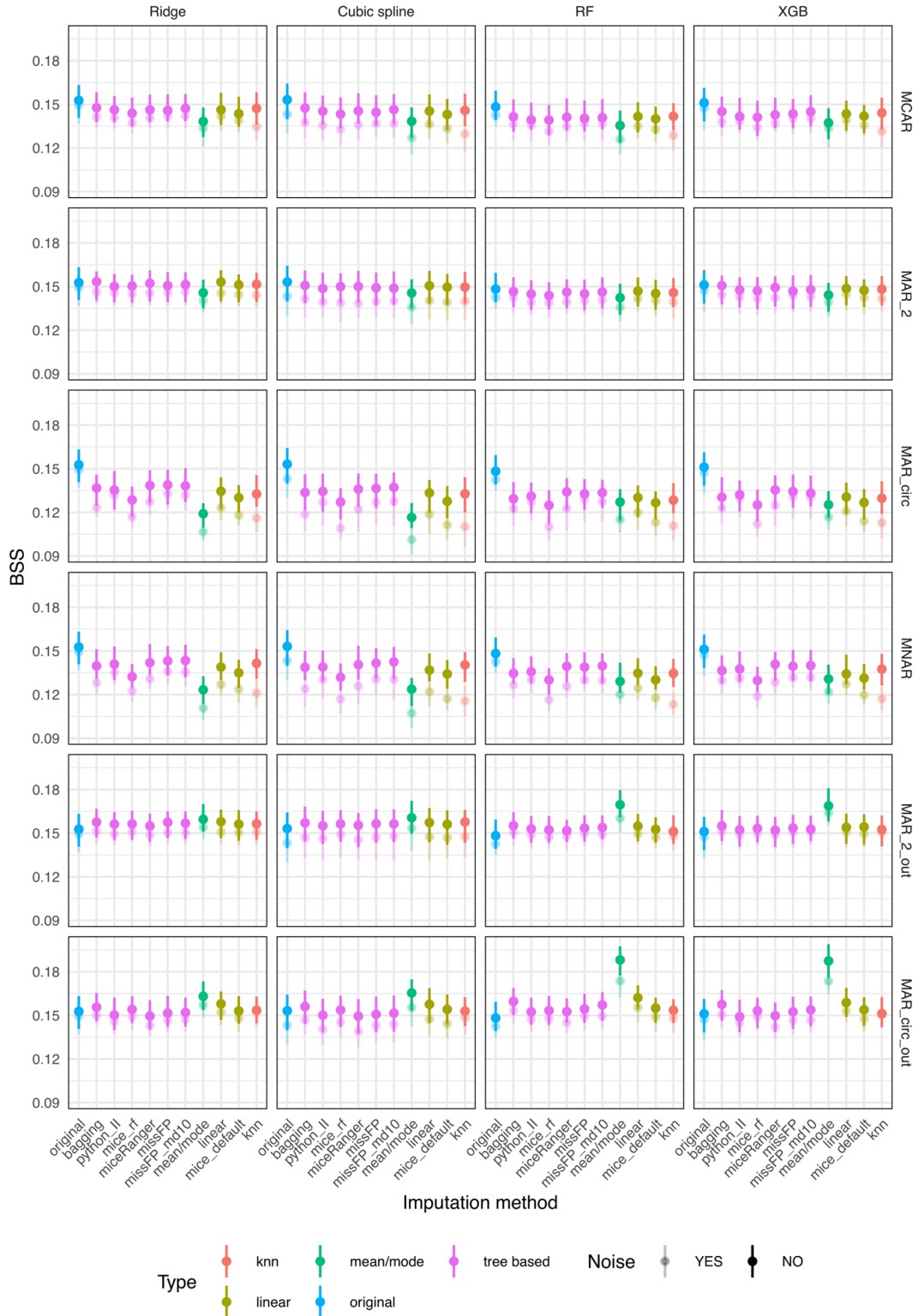

**Fig 7. Prediction performance (BSS) for simulated datasets with simulated missingness: high correlation (0.7) and low AUROC (0.75).**

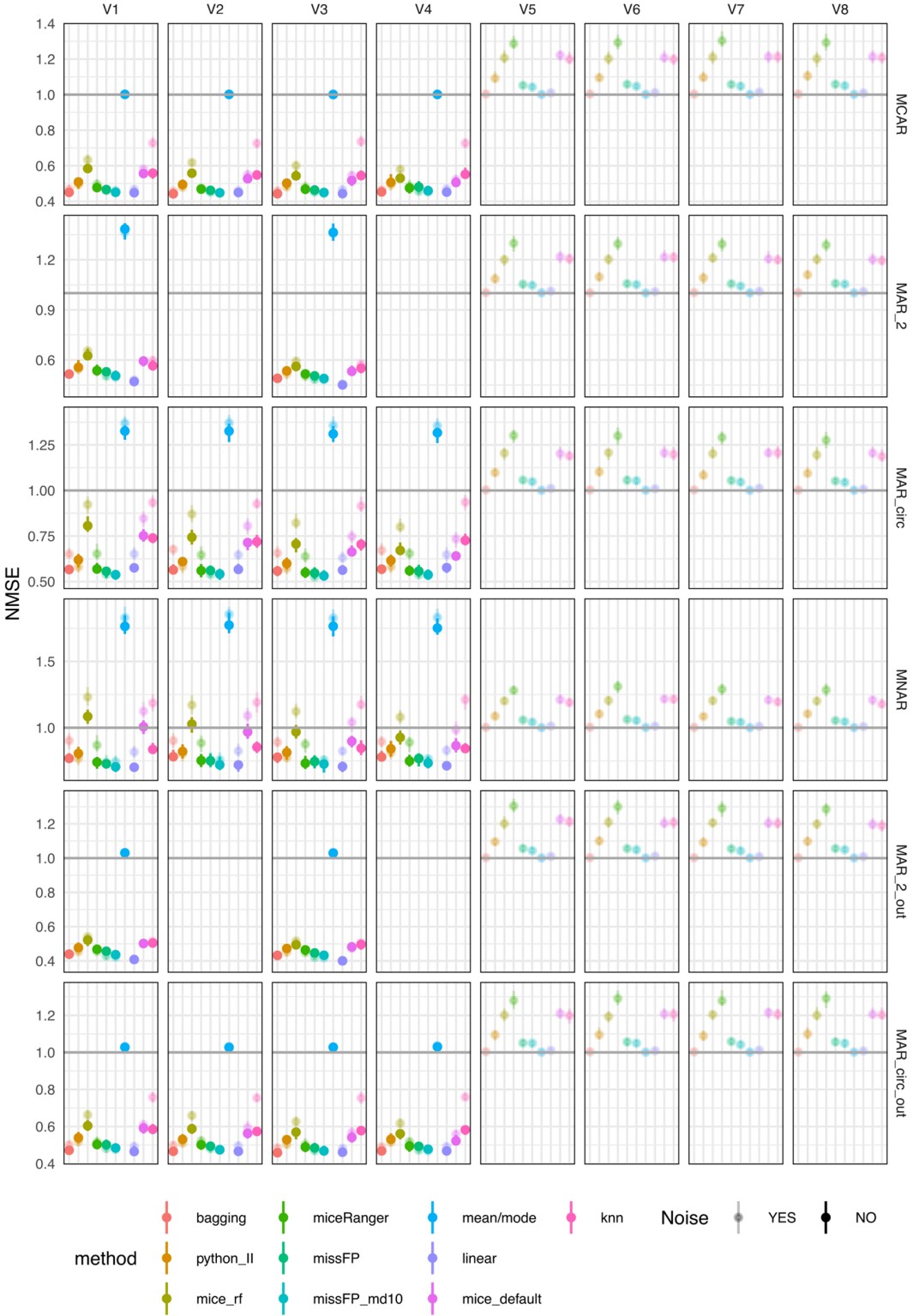

**Fig 8**. **NMSE errors (deviations from true values) on test sets for simulated datasets with simulated missingness: high correlation (0.7) and high AUROC (0.9).** To facilitate visualisation, only four out of the twelve noise variables are included in the figure.

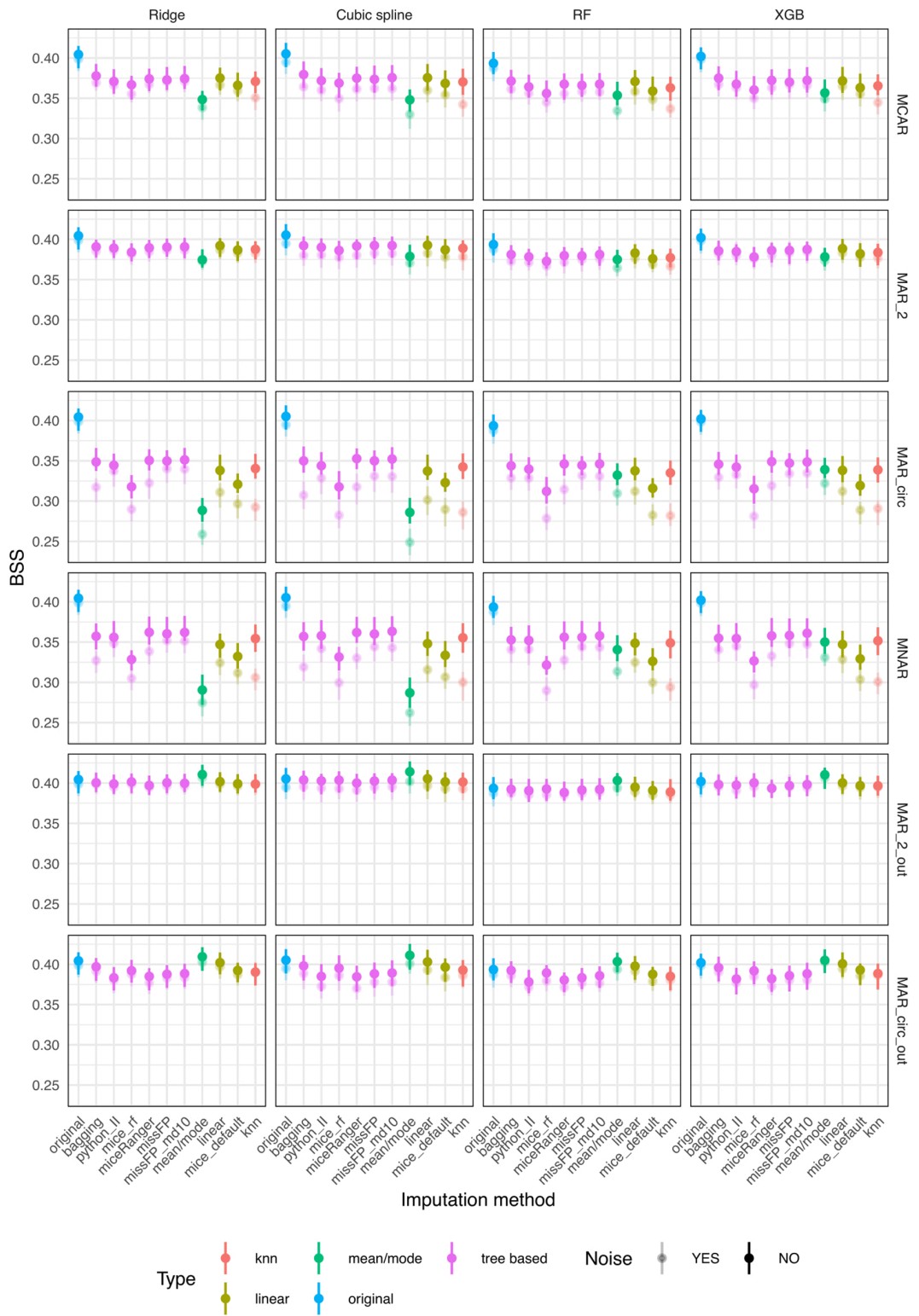

**Fig 9. Prediction performance (BSS) for simulated datasets with simulated missingness: high correlation (0.7) and high AUROC (0.9).**

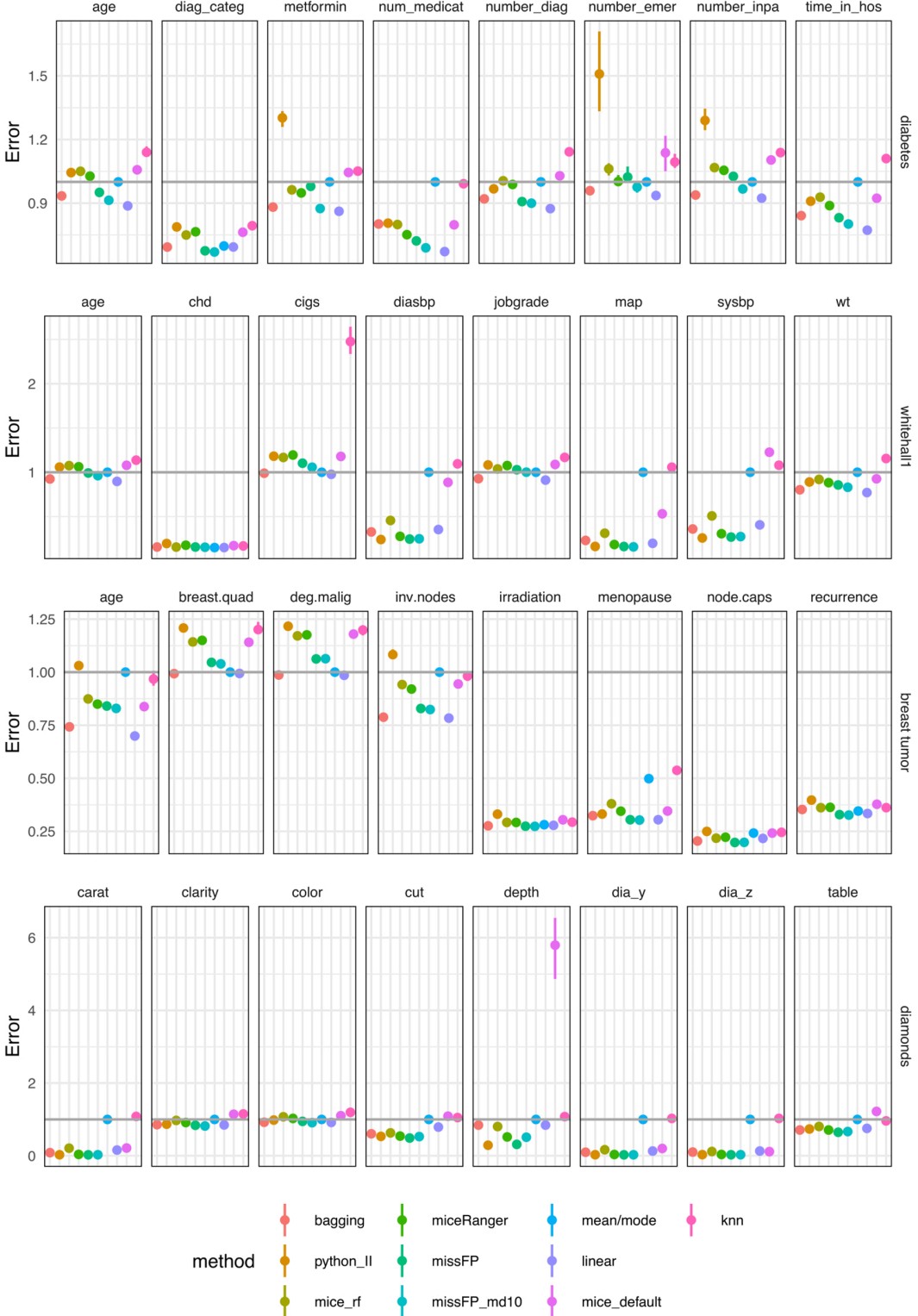

**Fig 10**. **Errors (deviations from true values); median and IQR bars.** For continuous variables the y-axis error is NMSE. For categorical variables the y-axis error is MER. The categorical variables are: diag_categ (diabetes dataset), irradiation, menopause, node.caps, recurrence (breast tumor

dataset); chd (whitehall1 dataset). All other variables are continuous. To facilitate visualization, the top eight most important variables for each dataset are selected; the ranking is done using the variable importance of the random forest model built on the original dataset (with no missing values).

prediction performance is measured by BSS for categorical outcomes and R squared for continuous outcomes (Fig 11). Bagging and linear imputation perform best on the datasets with categorical outcome; mean/mode, bagging, linear and missForestPredict perform best on the datasets with continuous outcome.

On the diabetes dataset, Iterative Imputer produces a variable-wise NMSE much greater than one on three variables: metformin, number_emergency, number_inpatient. kNN produces the largest NMSE, greater than one, on three other variables: age, number_diagnoses, time_in_hospital. mice (with default settings) also produces a high NMSE on number_inpatient and number_emergency which are the most important variables in prediction (variable importance is presented in the shiny app). Linear, bagging and missForestPredict (with shallow trees) imputation methods produce lowest NMSE, less than one on all variables. Iterative Imputer, mice (with default settings) and kNN yield the lowest BSS for all prediction models. Linear, bagging and missForestPredict (with shallow trees) produce the best performance results for all prediction models, although for RF the performance of all tree-based imputation methods (except Iterative Imputer) is very similar. Mean/mode imputation produces good performance results, better than mice (default), kNN and Iterative Imputer.

On the whitehall1 dataset kNN produces an NMSE much greater than one for the variable cigs and greater than one on diasbp and wt, on which the other methods produce NMSE less than one. mice (with default settings) produces an NMSE greater than one on sysbp on which all other methods (except kNN) produce an NMSE less than one. Linear, bagging and missForestPredict (with shallow trees) imputation methods generally produce lowest NMSE. kNN and mice (default) yield the lowest BSS for all prediction models. Linear and bagging produce the best performance results for all prediction models, followed by missForestPredict. Mean/mode imputation yields better results than mice (default) and kNN.

On the breast tumor dataset, linear and bagging imputation methods produce lowest NMSE on continuous variables, followed by missForestPredict. Mean/mode and kNN produce the highest MER on the categorical variable menopause. For RF and XGB prediction models, mean/mode, linear, bagging and missForestPredict perform best. For linear prediction models (Ridge and cubic splines), all tree-based imputation methods perform almost equally well, better than mean/mode, linear, mice (default) and kNN imputation methods.

On the diamonds dataset, mice (default) produces an extremely poor imputation for the variable depth with (median NMSE on the test set of 5.8). Mean/mode and kNN produce largest NMSE on variables carat, dia_y and dia_z, closer to or higher than one, while all other imputation methods yield low NMSE values. mice (default and RF) produce poor performance results for all prediction models. kNN and mean/mode also yield poor performance results for linear prediction models (Ridge and Cubic splines). Bagging, miceRanger and missForestPredict produce the best results for all prediction models.

The differences between missForestPredict with deep or shallow trees in terms of variable-wise performance and predictive performance are generally minimal for all datasets; on the diabetes dataset, missForestPredict with shallow trees seems to have a slight advantage. The OOB bias is similar except on diabetes dataset for which missForestPredict with shallow trees shows less bias on most variables (S1 Appendix). miceRanger produces smaller OOB bias on some continuous variables on diabetes and breast tumor datasets and higher bias on categorical variables on breast tumour dataset. missForestPredict with shallow trees also convergences faster on most datasets, with the largest difference on the diabetes dataset.

## Results on real datasets with missing values

Across all dataset-prediction model combinations, mean/mode, bagging and missForestPredict yield the best results (Fig 12), with some exceptions for the Ridge model. All datasets have categorical outcome.

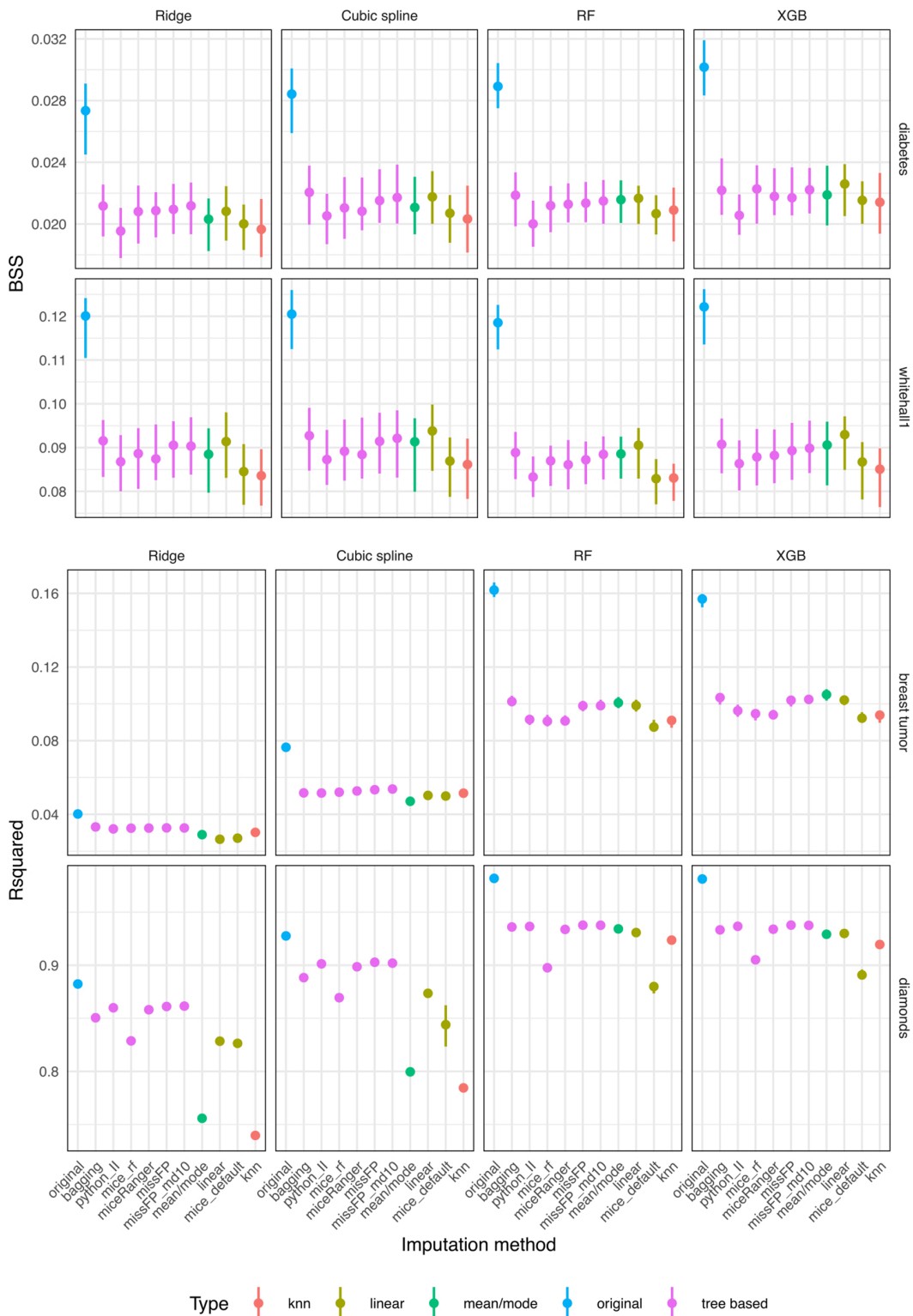

**Fig 11. Prediction performance for datasets with simulated missingness (BSS for categorical outcomes and R squared for continuous outcomes).** The Q1–Q3 intervals are small for R-squared and sometimes display as dots. The boxplots can be visualised in the shiny app.

As the deviation from true values (NMSE) cannot be calculated, because missingness was not simulated but present in these datasets, we present the distributions of continuous variables before and after imputation (Fig 13) and the proportion of values in each category before and after imputation (Fig 14), for categorical variables. We limit the results to the imputations on the first iteration and the top two variables in each dataset with highest missingness. Full results for all variables are available in S1 Appendix. Determining the desired distribution after imputation is difficult without knowing the true values. However, missForestPredict and Iterative Imputer produce values that more closely resemble the original distribution for ordinal variables in the diabetes dataset; these were treated as continuous variables across all imputation methods and predictive models.

The most striking differences can be observed on the covid dataset (three categorical variables with missing values). For cubic splines, RF and XGB models, mice (default), mice (RF) and miceRanger produce lower prediction performance. All other imputation methods perform similarly. For Ridge, on the contrary, mice (default), mice (RF) and miceRanger perform best. The BSS for Ridge models is though lower than the other models, even for the best imputation models, which can be explained by the serious miscalibration of all Ridge models (calibration results in shiny app); the mice methods seem to correct to some extend the miscalibration. missForestPredict converges fast, suggesting that the imputations do not diverge much from mode imputation (S1 Appendix).

On the CRASH-2 dataset, mean/mode and bagging perform best for RF and XGB models. For regression models (Ridge and cubic splines), the differences are small. On the diabetes dataset only one categorical variable is in top eight most important variables and the differences between imputation methods are very small.

On the IST dataset, the differences in BSS are small (0.001 or less) for all models except the Ridge model, for which linear imputation and Iterative Imputer perform best (BSS of 0.125) and mean/mode performs worst (BSS of 0.103). mice (default) has not been run on the IST dataset due to errors because of a rank deficient matrix ("error in chol.default(): ! the leading minor of order 23 is not positive definite").

The differences in prediction performance between missForestPredict with deep or shallow trees are very small for all datasets. The number of iterations until convergence is also similar (S1 Appendix).

## Computational efficiency and memory utilization

The median runtimes for: (1) building the imputation model and imputing the train set and (2) imputing the test set are presented in Table 9. For some methods, imputing the train set is part of the model building, therefore we present the imputation model building time together with the time for imputing the train set. We exclude the time for mean/mode imputation, which is almost zero. Only a subset of the simulated datasets is included for MNAR missingness mechanism. Full results (median and IQR) for all datasets, including separate imputation model building runtimes are available in the shiny app.

Linear imputation is generally the fastest imputation method for fitting the imputation models and imputing the train dataset. On the smallest datasets (simulated datasets and covid dataset), missForestPredict is as fast or comparable to the linear imputation, with runtimes under one second. For the datasets with low correlation (sim_90_1 and sim_90_1_noise), missForestPredict with deep trees runs faster than missForestPredict with shallow trees because it converges after one iteration (S1 Appendix). kNN is also fast on small datasets. On larger datasets, missForestPredict (with deep or shallow trees) remains the second fastest after the linear imputation. missForestPredict with shallow trees provides a computation advantage on large datasets, with the largest difference on the diabetes datasets, where using deep trees produces runtimes of almost 3 times the runtime of missForestPredict with shallow trees. kNN is though much slower on the largest datasets. Despite the fact that miceRanger uses multiple imputation and performs predictive mean matching, but owing to the parallelization implemented in both ranger and the miceRanger package, it produces competitive runtimes for model fitting and imputing the train set, comparable to Iterative Imputer and much faster the mice (RF) and mice (default), which are the slowest overall.

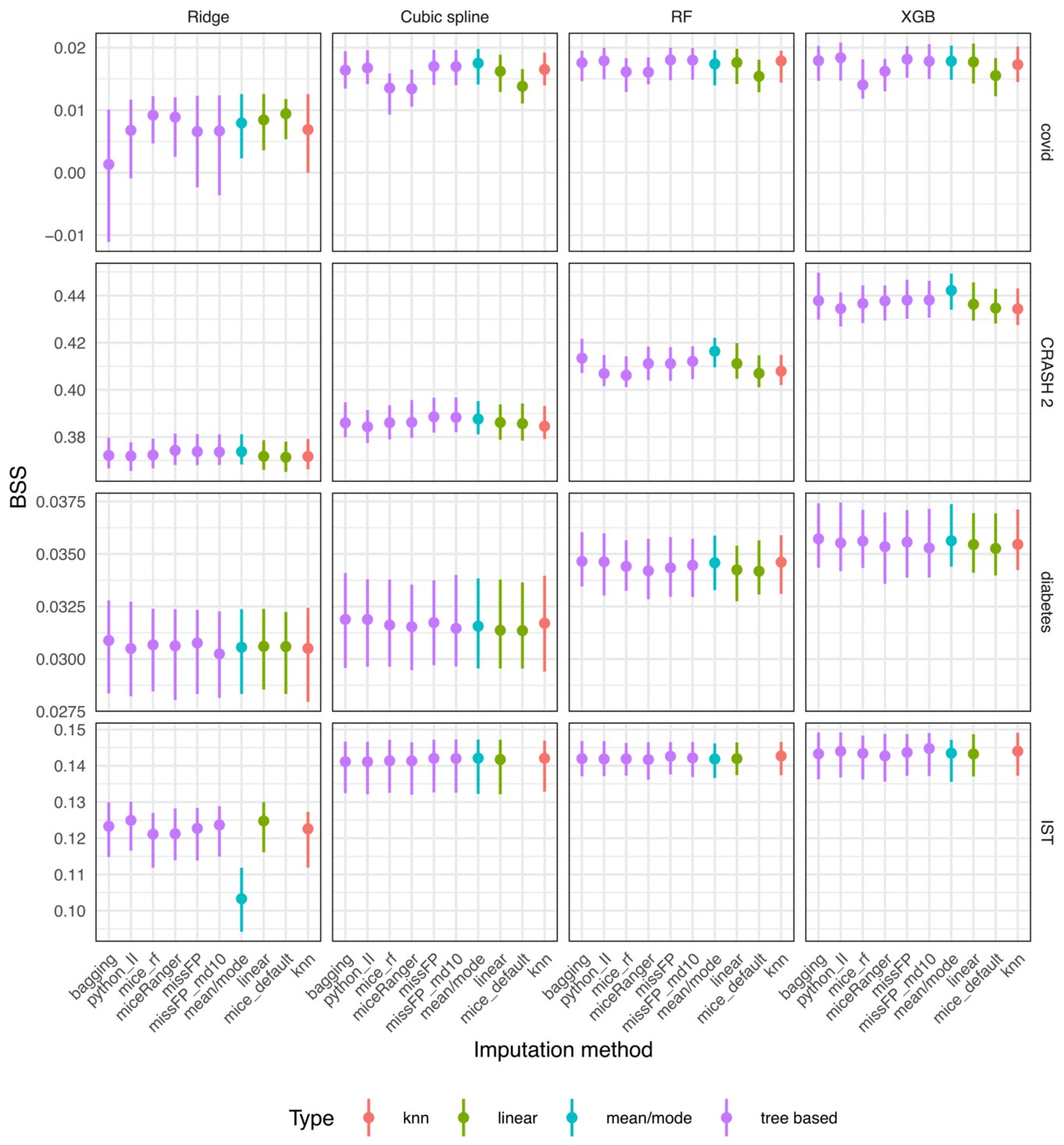

**Fig 12. Prediction performance for datasets with missing values and categorical outcome (BSS).**

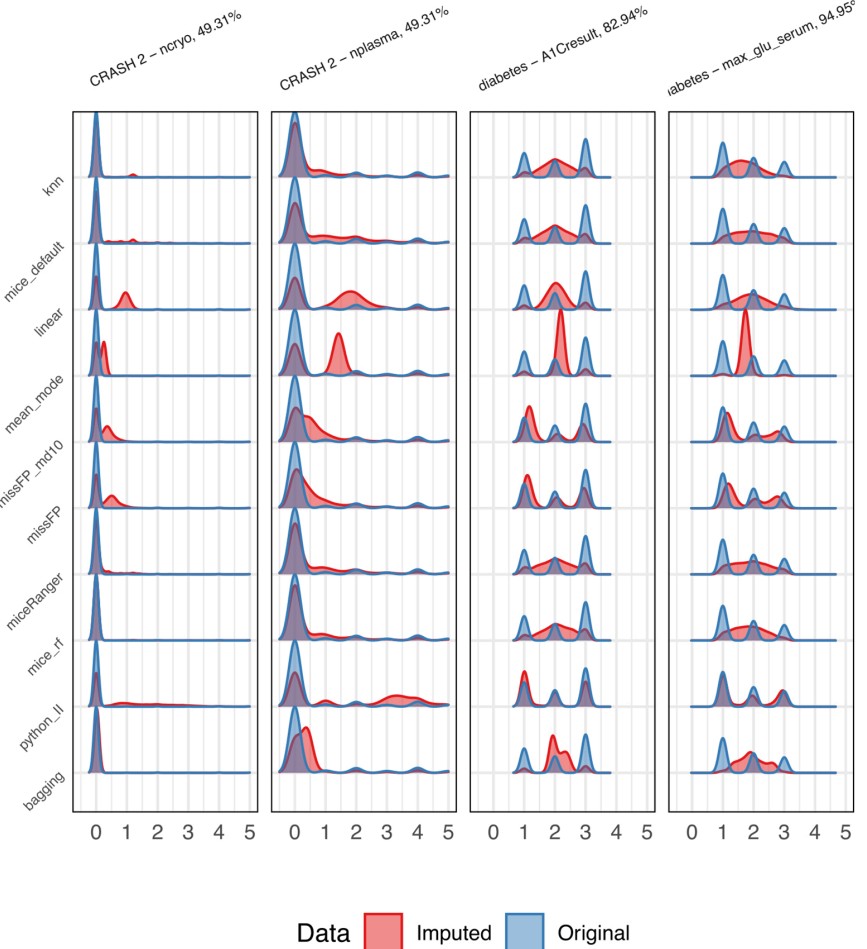

**Fig 13. Distribution of original (non-missing) and imputed values for continuous variables.** The x-axis has been limited to a value of 5 for better visualisation.

When imputing new data (test set), kNN is the slowest when neighbours have to be found in a large dataset (e.g.: breast tumour, diamonds, diabetes). missForestPredict provides fast imputations, comparable to linear imputation, with shallow trees showing an advantage on the large datasets. mice (RF) and miceRanger provide slower imputation times, but faster than kNN on large datasets.

As most methods build and store in memory models for imputation, the memory requirements on large datasets depend on the size of the imputation models. As for runtimes, we show a limited selection the simulated datasets as the interest in memory utilization is mostly for large datasets (Table 10). mice (default and RF) and kNN do not store models, therefore the imputation model object sizes are rather small. Linear models also have small object sizes and implicitly low memory requirements. The tree-based imputation methods (other than mice RF) - which have all been set to use 100 trees - require more memory, with bagging being the most memory-intensive, followed by missForestPredict with deep trees. missForestPredict with shallow trees shows a considerable reduction in memory usage on large datasets (e.g. more than 10-fold reduction on diabetes dataset), and the object sizes are lower than miceRanger, making it the least memory-intensive method among the tree-based imputation methods that rely on stored imputation models.

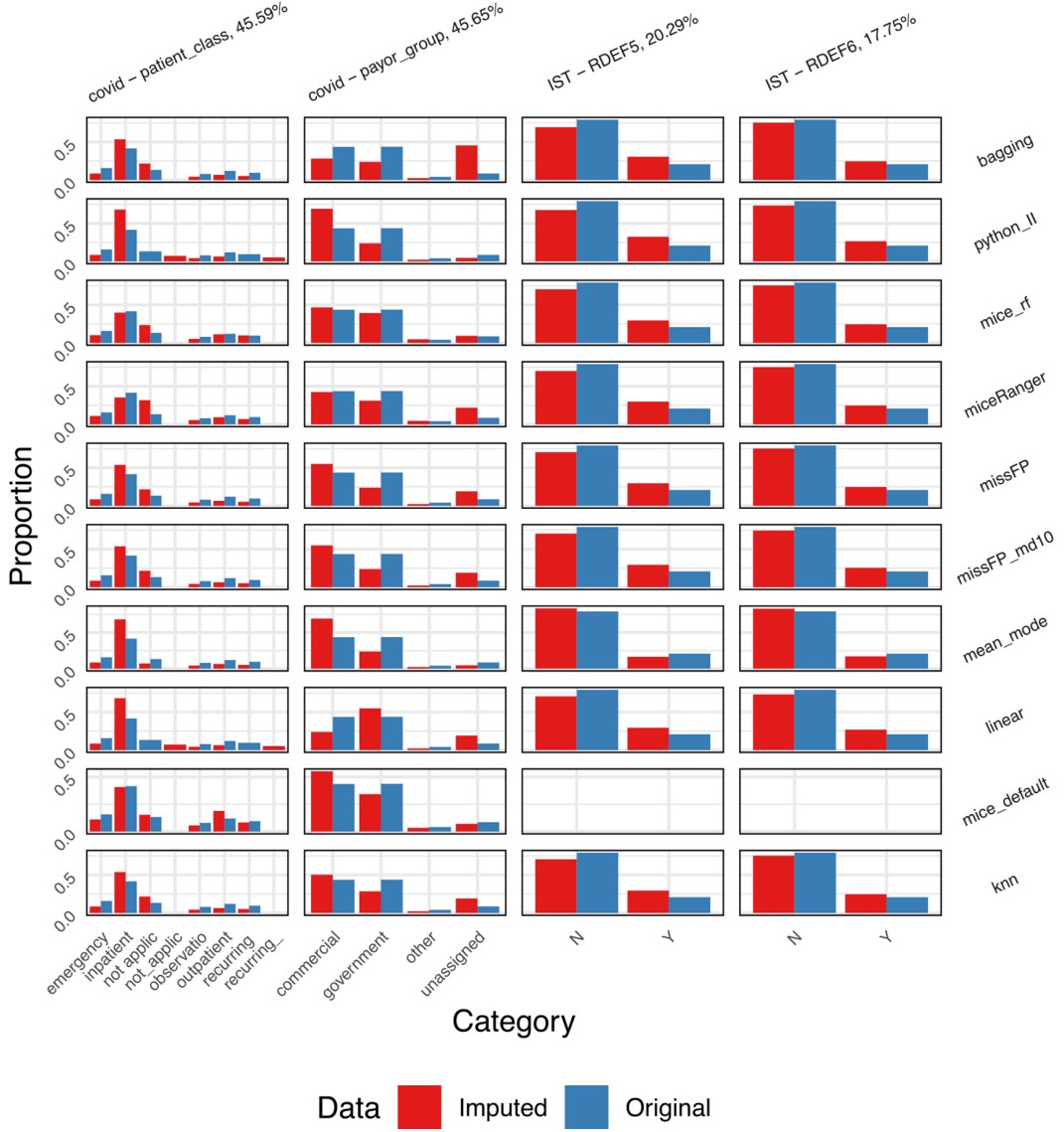

**Fig 14**. **Proportion of original (non-missing) and imputed values for categorical variables.**

## Discussion

We introduced missForestPredict algorithm in the current landscape of imputation methods that can be applied when data are expected to be missing at prediction time. The missForestPredict package provides single imputation in an iterative manner, until a convergence criterion (based on OOB NMSE) is reached or for a fixed number of iterations. It offers control on the variables to impute and the variables used as predictors for imputation. The OOB errors for each variable at each iteration can be examined post imputation.

We compared the performance of missForestPredict to RF based imputation methods in R and python, and included some other popular imputation methods: mean/mode imputation, linear imputation, mice and kNN. We performed the comparisons on eight simulated datasets with six missing data amputation methods (48 scenarios) and on eight large real datasets and evaluated the prediction performance (BSS and R squared) of four prediction models (two linear and two

**Table 9. Median runtime in seconds for building imputation models and imputing the training set (the Q1 - Q3 range is presented in the shiny app).**

| dataset | type | linear | mice_default | knn | bagging | python_II | mice_rf | miceRanger | missFP | missFP_md10 |
|---|---|---|---|---|---|---|---|---|---|---|
| sim_90_1 - MNAR | Train (fit + impute) | 0.1 | 12 | 0.4 | 2.2 | 6.6 | 19.7 | 13.3 | 0.1 | 0.4 |
| sim_90_1_noise - MNAR | Train (fit + impute) | 0.7 | 13.3 | 4.3 | 27.1 | 33.6 | 44.5 | 15.7 | 0.8 | 2.6 |
| sim_90_7 - MNAR | Train (fit + impute) | 0.1 | 12.2 | 0.4 | 1.9 | 6.7 | 19.8 | 14 | 0.5 | 0.3 |
| sim_90_7_noise - MNAR | Train (fit + impute) | 0.7 | 13.1 | 4.3 | 26.1 | 33.8 | 44.2 | 15.7 | 4 | 3 |
| breast tumor | Train (fit + impute) | 0.6 | 551.5 | 250.5 | 397 | 40.8 | 774.4 | 43.4 | 24.6 | 17.5 |
| diamonds | Train (fit + impute) | 1 | 254.6 | 225.9 | 78.5 | 57.7 | 272.5 | 57.6 | 23.8 | 11.4 |
| diabetes | Train (fit + impute) | 3.2 | 7800.5 | 1042.3 | 677.8 | 163.7 | 1395.9 | 216.3 | 56.1 | 18.7 |
| whitehall1 | Train (fit + impute) | 0.5 | 15.9 | 28.2 | 39.5 | 26.8 | 112.3 | 24.6 | 9.9 | 5.7 |
| CRASH 2 - MV | Train (fit + impute) | 1.9 | 12395.6 | 51.6 | 211.6 | 78.2 | 298.1 | 44.6 | 17.8 | 11.5 |
| diabetes - MV | Train (fit + impute) | 1.9 | 3167.6 | 155.6 | 291.7 | 203 | 289.7 | 146.4 | 25 | 15.2 |
| covid - MV | Train (fit + impute) | 0.5 | 41.4 | 4.4 | 5.5 | 23.4 | 79 | 12.2 | 0.8 | 0.8 |
| IST - MV | Train (fit + impute) | 1.3 | NA | 24.8 | 95.6 | 96.5 | 244.3 | 25.1 | 11.1 | 7 |
| sim_90_1 - MNAR | Test (impute) | 0 | 0.1 | 0.2 | 0.1 | 0.7 | 1.1 | 2.1 | 0 | 0.1 |
| sim_90_1_noise - MNAR | Test (impute) | 0.1 | 2 | 2.2 | 0.2 | 2.9 | 5.6 | 8.8 | 0 | 0.3 |
| sim_90_7 - MNAR | Test (impute) | 0 | 0.1 | 0.2 | 0.1 | 0.7 | 1.1 | 2.3 | 0.1 | 0 |
| sim_90_7_noise - MNAR | Test (impute) | 0.1 | 2 | 2.2 | 0.2 | 2.8 | 5.5 | 8.5 | 0.4 | 0.3 |
| breast tumor | Test (impute) | 0.1 | 12.8 | 125.2 | 121.4 | 3 | 123.3 | 47.5 | 2.6 | 1.5 |
| diamonds | Test (impute) | 0.2 | 4.6 | 112.9 | 0.6 | 3.1 | 36.2 | 84.8 | 2.7 | 0.8 |
| diabetes | Test (impute) | 0.4 | 59.3 | 520.8 | 117.4 | 8.6 | 217.4 | 158.2 | 5.5 | 1.2 |
| whitehall1 | Test (impute) | 0.1 | 2.2 | 14.1 | 2.8 | 2.8 | 14.2 | 17 | 1.1 | 0.5 |
| CRASH 2 - MV | Test (impute) | 0.2 | 24.5 | 25.9 | 3.5 | 4.5 | 38.4 | 32.7 | 1.2 | 0.5 |
| diabetes - MV | Test (impute) | 0.2 | 88.1 | 77.7 | 36.8 | 3.4 | 49.8 | 52.3 | 3.2 | 1.4 |
| covid - MV | Test (impute) | 0.1 | 5.3 | 2.2 | 0.2 | 2.2 | 12.9 | 1.4 | 0 | 0 |
| IST - MV | Test (impute) | 0.1 | NA | 12.3 | 4.5 | 2.1 | 35.5 | 4.8 | 1 | 0.4 |

**Table 10. Median object size of the imputation object in MB (the Q1 - Q3 range is presented in the shiny app).**

| dataset | linear | mice_default | knn | bagging | mice_rf | miceRanger | missFP | missFP_md10 |
|---|---|---|---|---|---|---|---|---|
| sim_90_1 - MNAR | 0.5 | 0.6 | 0.2 | 274.3 | 0.6 | 68.5 | 14.1 | 28.3 |
| sim_90_1_noise - MNAR | 6.4 | 2 | 0.7 | 1528.9 | 2 | 291.7 | 63.5 | 95.2 |
| sim_90_7 - MNAR | 0.5 | 0.6 | 0.2 | 272.9 | 0.6 | 79.1 | 59.6 | 26.6 |
| sim_90_7_noise - MNAR | 6.4 | 2.1 | 0.7 | 1530.5 | 2.1 | 292 | 314.1 | 113.5 |
| breast tumor | 32.1 | 14.8 | 3.6 | 8857.7 | 14.8 | 372.1 | 870.2 | 277.8 |
| diamonds | 24.3 | 15.7 | 5 | 9311.6 | 15.7 | 375.2 | 2759.4 | 220.3 |
| diabetes | 205.8 | 29.8 | 9.8 | 25686.3 | 29.8 | 964.5 | 3439.1 | 241 |
| whitehall1 | 9.8 | 5.5 | 1.7 | 3368.5 | 5.5 | 290.1 | 1003.5 | 186.1 |
| CRASH 2 - MV | 76.7 | 7.6 | 4.4 | 12095.3 | 7.6 | 912 | 1278.5 | 255.4 |
| diabetes - MV | 125.1 | 19.2 | 11.7 | 8433.1 | 19.2 | 411.6 | 1680.7 | 241.2 |
| covid - MV | 13 | 2 | 0.5 | 573.3 | 2 | 50 | 5.1 | 5.1 |
| IST - MV | 53.8 | NA | 3.1 | 4412.7 | 4.6 | 466.6 | 606.9 | 159.5 |

tree-based) and the deviation from true values for simulated missing data scenarios. We share our code and allow other researchers to easily run the comparison on new datasets by following simple instructions. missForestPredict provides among the best results in most of the scenarios, although the differences between most tree-based imputation methods are generally small in most scenarios.

In low correlation settings and outcome independent missing data mechanism, mice (default and RF), kNN, miceRanger and Iterative Imputer deviate heavily from true values (NMSE greater than one) and produce inferior prediction performance

results, suggesting that these imputation methods tend to "over-impute". These are "low imputability" settings, that is: other variables do not contain enough predictive information to impute a specific variable, making it hard for imputation methods to provide imputations better than mean imputation. Linear imputation, bagging and missForestPredict with deep trees have superior predictive performance, especially in the high AUROC scenarios. missForestPredict with shallow trees tends though to slightly over-impute, although to a lesser extent than the other methods. missForestPredict with deep trees mostly converges after one iteration and provides imputations close to mean imputation, while missForest-Predict with shallow trees runs for multiple iterations. In the absence of other knowledge about the data at hand, mean imputation is a reasonable choice in such situations. Sophisticated methods might overfit in low imputability settings. On real datasets with missing values it is though difficult to know a priori if there is enough information in other variables to impute a specific variable, as the correlation structure between variables is more complex than in our simulated settings. A heuristic that though can be used is inspecting the OOB NMSE on each variable for the first iteration of missForestPre-dict. If OOB NMSE is close to one (or even greater than one) for some variables, this can be used as an indication of low imputability of such variables. We advise one iteration because we expect the OOB NMSE to be less biased at first itera-tion than at later iterations (when imputed values of one variable are reinforced by imputed values of another feature that in turn was imputed based on the first variable). By inspecting the OOB NMSE presented in the S1 Appendix, this strategy seems to work in all scenarios, including MNAR and outcome related MAR. The difference in convergence between miss-ForestPredict with deep and shallow trees can be explained by the tendency of deep forests to memorize training data (isolate observations in final leaves of the trees), which leads to good training performance but low (not better than mean) performance on unseen data. This behaviour is exemplified in the work of Barreñada et al [22]. As missForestPredict uses the OOB observations for calculating the convergence criteria, which can be considered "unseen" in the training process, the OOB NMSE will show poor performance, not better than mean, and the algorithm will converge after first iteration.

In high correlation settings and outcome independent missing data mechanism, all imputation methods produce smaller deviations from true values when compared to mean/mode. Mean/mode is no longer an optimal choice for linear pre-diction models (Ridge and cubic splines), but performs relatively well for tree based models (RF and XGB). This can be explained, in theory, by the fact that the mean imputed observations could be isolated at a split in a tree, followed by a split of that node on a different variable with less imputed values, similar for missing incorporated in attribute [23] or miss-ing value indicators. We provide no proof though that this is the mechanism that explains the relatively good performance of mean/mode imputation for RF and XGB prediction models. missForestPredict, bagging and miceRanger provide the best results across al scenarios in these "high imputability" settings, even for linear prediction models. Even if our sim-ulation scenarios were tailored for regression models, as we did not explicitly include complex association structures between variables, these methods provide slightly superior results in some scenarios when compared to linear imputation or kNN. For tree-based prediction models (RF and XGB), mice (default and RF) produces inferior results compared to all other methods, including mean/mode; for linear prediction models (Ridge and cubic splines) they produce better results than mean/mode, but worse than all other methods. kNN and mice (default and RF) also seem more susceptible to noise in these settings.

When missingness is outcome related, the amputation process adds information to the data structure. When AUROC is low, the information gain due to predictive missingness is higher relative to the predictive information contained in the vari-ables. Mean imputation exceeds the performance of the original datasets without missing values; mean imputation acts here as "constant value imputation" and its signal information is particularly well picked by tree based models which could, in principle isolate the observations imputed with a constant value by splits left and right to the constant value and isolate their relation the outcome. This behaviour of tree based models has been previously mentioned in the work of Josse et al [24]. Linear models can also pick this information, but to a lesser extent. A better comparison baseline than the origi-nal dataset without missing values would have been a prediction model with missing value indicators (MVIs), reflecting the information added by the missing data generation process. We did though not intend to study the impact of MVIs and focused only on the imputation methods. Linear and bagging imputation perform better as they probably do not deviate

considerably from mean. In low correlation settings, missForestPredict with deep trees also acts as a simple imputation method, as it converges fast. In high correlation settings though, missForestPredict will rather impute values closer to the true values, deviating from the mean and providing less performant results. One attention point for outcome related missingness and use of mean (or constant value imputation) together with tree-based prediction models (RF, XGB) is that these models will make use of it, acting similar to a missing value indicator. There might be situations when this is not desired, for example if the process of recording the data (and therefore of the missingness mechanism) is expected to change in future data (dataset shift on MVIs), but the variables (with their true values, known or missing) are expected to remain stationary. In this case, more sophisticated imputation methods that impute values closer to the true value might be preferable.

Although the open-access public datasets with missing values that allow research to occur in settings closer to reality are scarce, we have found four such datasets meeting our inclusion criteria (more than 10000 observations and more than 200 events in test sets for binary outcomes). Linear, bagging and missForestPredict imputation generally perform well on these datasets and mice (default and RF) generally underperforms. On the real datasets with MCAR simulated missingness, missForestPredict and bagging provide the best results for most prediction models. Linear imputation also performs well, even in the presence of categorical variables, for which the method is not specifically tailored because logistic regression imputation is not implemented in tidymodels.

On these large datasets, missForestPredict provides the fastest imputation after linear imputation, both for learning the imputation models and for imputing new observations. Among the imputation methods that internally store the imputation models, missForestPredict is the least memory-intensive. Setting the max depth of trees to 10 comes with both a computational and a memory usage advantage on large datasets, without compromising the predictive performance. At the other extreme, mice (default and RF) is notoriously slow on large datasets, but the least memory intensive, as it does not internally store models, and bagging is the most memory-intensive.

Our comparison study has a number of limitations. We studied a limited number of algorithms and options for imputation. We included only impute-then-predict scenarios, without focusing on models that incorporate the missingness in the model building process (as do random forests with surrogate splits or missing incorporated in attribute). We also did not study the impact of missing value indicators, alone or together with imputation methods, although they can be of predictive value [25]. Our simulated data scenarios are simplistic and aimed to provide a first base for observing the behaviour of different imputation methods. For example, it is unlikely that in real data four variables have the same pairwise correlation and equal importance in prediction. We included a limited number of real datasets, selecting them based on stringent criteria in terms of dataset size. It has been shown before that mean imputation is consistent when missing values are not informative [24], which might explain the good performance of mean imputation on large datasets. Smaller datasets (but still large enough to accommodate for machine learning prediction models) could be further studied. Additionally, the datasets used in our study included at most 25 variables with missing values. Future research could explore datasets with a larger number of variables with missing values, for example in the order of hundreds, to investigate whether the max_depth adjustment is sufficient for memory requirements in such settings.

All imputation methods we compared have been used with the default settings of the corresponding R or python library, with the exception that we have set the number of trees for all tree based methods to 100. kNN might benefit from tuning the number of neighbours, while we used the default of 5. For the iterative methods we have used the default number of iterations, while they might benefit from adjusting it (e.g.: for miceRanger we used the default of 5, for Iterative Imputer the default of 10, and for mice the default of 5 at training time and one at prediction time). Bagging outperforms many other imputation methods in many scenarios, despite being the simplest tree based imputation method: it does not provide multiple imputation, it is not iterative and does not use predictive mean matching. In datasets with a limited number of variables, it is expected that one or few (if any) of the variables are predictive for imputing a variable with missing values. Setting the number of variables to consider for splitting at each node to a large value might be beneficial for all tree based imputation methods. Despite its ability to produce unbiased results and correctly estimate the standard errors of estimates

in statistical inference settings mice underperforms consistently across scenarios, both in terms of deviation from true values and predictive performance. miceRanger performs multiple imputation but it does not outperform missForestPredict, despite the iterative algorithm being essentially the same. Linear imputation, as implemented in the tidymodels package and without native support for categorical variables, performs effectively in simple settings, but when the data structure is more complex (e.g: diamonds dataset), it underperforms compared to tree based methods. A predictor matrix could be handcrafted in such situations, including interactions and non-linearities in the imputation models. Considering that missForestPredict is rather fast even on large datasets, and it natively handles the data complexities, it is difficult to justify the need for handcrafting complex linear imputation models.

There is no shortage of studies comparing missing data imputation methods, and the original missForest algorithm has been repeatedly assessed as an algorithm that provides imputations close to the true values [26,27], and demonstrated good performance in prediction models using train/test split after imputing the entire dataset [28] or imputing the test set separately [29]. However, studies focusing on imputation at prediction time, as done in real applications are scarce [23, 24]. In the light of the current findings, we hope to set directions for further research to address several questions we left unanswered. Is iterative imputation beneficial or would one iteration (as in bagging) suffice for most methods? Is predictive mean matching useful in prediction applications? For methods that use an initialization scheme (mice, miceRanger, missForestPredict), which is either random selection from observed values or mean/mode initialization, would changing the initialization scheme change the results? For tree-based methods, is the number of trees or the number of variables tried at each split impacting the imputation or predictive performance? In which situations does the variable-wise imputation error correlate to the predictive performance? Is multiple imputation necessary in prediction settings? Would confidence intervals of predictive measures for the final prediction model be smaller when multiple imputation is used? Would averaging predictions of multiple models built on imputed datasets, instead averaging imputations as done in the current study show a benefit? The missForestPredict package provides single imputation in an iterative fashion. Nevertheless, multiple imputation using the same algorithm can be performed by repeating the imputation a number of times. Previous research [30] suggests no benefit of multiple imputation in predition settings, although their imputation models are not applied to new observations, but rather rebuilt on the test set. The factors that we expect to impact the final prediction performance of an imputation strategy would be: the importance of variables in the prediction model; their missingness rate; the missingness pattern; and the information contained in other variables when imputing (important) variables (the "imputability" of variables). Simulation scenarios could include different rates of missingness in important variables, different missingness patterns (e.g.: variables missing together) and complex data structures with regards to the association between variables. Additionally, we investigated the impact of "noise" variables on simulated datasets, by adding twelve noise variables to datasets with four "signal" variables, resulting in a high noise to signal ratio of 3:1. In all scenarios, the addition of noise variables negatively affected on the final performance of the prediction model, which can be due its impact on the imputations, its direct impact on the prediction model, or both. Further research could explore different levels of noise and disentangle the performance decrease attributable to imputation from that caused by the prediction model itself. We hope future research will shed light on these important topics for prediction settings when missing data are expected at prediction time.

## Conclusion

The missForestPredict R package implements iterative imputation using random forests and is tailored for prediction settings when data are expected to be missing at prediction time. It is user friendly and offers additional functionality, like extended error monitoring, control over variables used in the imputation and custom initialization allowing users the flexibility to adapt the algorithm to their settings. missForestPredict provides competitive imputation results while remaining both time-efficient and memory-efficient. In our comparison study, we observe that some complex imputation methods (kNN, Iterative Imputer, mice, miceRanger) lead to worse predictive performance than mean imputation when there

are low correlations between variables. RF based imputation methods perform better in scenarios with high correlations between variables. missForestPredict performs well in both scenarios, although the differences between methods are generally small.

## Acknowledgments

We warmly thank Lars Bosshard and Daniel Stekhoven for the insightful discussions during the early stages of developing the missForestPredict package and for their review of the initial version of the Methods section in this study. The resources and services used in this work were provided by the VSC (Flemish Supercomputer Center), funded by the Research Foundation - Flanders (FWO) and the Flemish Government.

## Supporting information

**S1 Appendix. The file contains supporting information for methods, datasets descriptions and additional results.** (PDF)

## Author contributions

**Conceptualization:** Elena Albu, Laure Wynants, Ben Van Calster.

**Data curation:** Elena Albu.

**Formal analysis:** Elena Albu, Shan Gao.

**Funding acquisition:** Ben Van Calster.

**Methodology:** Elena Albu, Shan Gao, Laure Wynants, Ben Van Calster.

**Project administration:** Laure Wynants, Ben Van Calster.

**Resources:** Ben Van Calster.

**Software:** Elena Albu, Shan Gao.

**Supervision:** Laure Wynants, Ben Van Calster.

**Validation:** Elena Albu.

**Visualization:** Elena Albu.

**Writing – original draft:** Elena Albu.

**Writing – review & editing:** Elena Albu, Shan Gao, Laure Wynants, Ben Van Calster.

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
