## [Decision Letter · Decision Letter 0]

27 Nov 2024

PONE-D-24-33134missForestPredict – Missing data imputation for prediction settingsPLOS ONE

Dear Dr. Albu,

Thank you for submitting your manuscript to PLOS ONE. After careful consideration, we feel that it has merit but does not fully meet PLOS ONE’s publication criteria as it currently stands. Therefore, we invite you to submit a revised version of the manuscript that addresses the points raised during the review process.

We look forward to receiving your revised manuscript.

Kind regards,

Santiago Callegari, MD

Academic Editor

PLOS ONE

 Journal requirements: When submitting your revision, we need you to address these additional requirements. 1. Please ensure that your manuscript meets PLOS ONE's style requirements, including those for file naming. The PLOS ONE style templates can be found at https://journals.plos.org/plosone/s/file?id=wjVg/PLOSOne_formatting_sample_main_body.pdf and https://journals.plos.org/plosone/s/file?id=ba62/PLOSOne_formatting_sample_title_authors_affiliations.pdf 2. We note that the grant information you provided in the ‘Funding Information’ and ‘Financial Disclosure’ sections do not match.  When you resubmit, please ensure that you provide the correct grant numbers for the awards you received for your study in the ‘Funding Information’ section. 3. Please include captions for your Supporting Information files at the end of your manuscript, and update any in-text citations to match accordingly. Please see our Supporting Information guidelines for more information: http://journals.plos.org/plosone/s/supporting-information. 

Reviewers' comments:

Reviewer's Responses to Questions

**Comments to the Author**

1. Is the manuscript technically sound, and do the data support the conclusions?

Reviewer #1: Yes

2. Has the statistical analysis been performed appropriately and rigorously?

Reviewer #1: Yes

3. Have the authors made all data underlying the findings in their manuscript fully available?

Reviewer #1: Yes

4. Is the manuscript presented in an intelligible fashion and written in standard English?

Reviewer #1: Yes

5. Review Comments to the Author

Reviewer #1: In this manuscript, the authors present a new package that offers the convenience of random forest powered imputation coupled with prediction. The material presented is robust and complete. The R package is actually already on CRAN, so I think this should complement nicely. I don't think people would wish for more.

I have very few concerns, and I feel this manuscript should be made public without delay.

- Although I realize it's something of an extension of missForest, but it seems like there's a distinct lack of any consideration of the results of the imputation. It's so focused on predictions, although it's only half the process. The manuscript is so complete except for any visualization or analysis of what is imputed. I wouldn't do more than a figure or two. Then readers would have everything in one place.

- In figures 2,3,4,6, and 8 the y axis is labeled "value", perhaps that could be renamed "Error" or "NMSE"?

- Seems like linear and mean imputation is often times the biggest competitor, could consider moving missForestPredict next to them in the plots, because that's what I was always trying to compare... looking across the rows of dots.

- Line 169, should that p_obs be a subscript "obs"?

- I wonder if a lighter application of noise wouldn't have been better... had a huge effect.

- Nice code!

6. PLOS authors have the option to publish the peer review history of their article (what does this mean?). If published, this will include your full peer review and any attached files.

Reviewer #1: No

---

## [Author Response · Author response to Decision Letter 1]

30 Jan 2025

We sincerely appreciate the reviewer's time and effort in carefully reading our manuscript and providing attentive feedback. We have addressed each comment in detail in the "Response to Reviewers" document.

Thank you for your valuable insights!

---

## [Decision Letter · Decision Letter 1]

5 May 2025

PONE-D-24-33134R1missForestPredict – Missing data imputation for prediction settingsPLOS ONE

Dear Dr. Albu,

Thank you for submitting your manuscript to PLOS ONE. After careful consideration, we feel that it has merit but does not fully meet PLOS ONE’s publication criteria as it currently stands. Therefore, we invite you to submit a revised version of the manuscript that addresses the points raised during the review process.

We look forward to receiving your revised manuscript.

Kind regards,

Leona Cilar Budler

Academic Editor

PLOS ONE

Additional Editor Comments:

Dear authors,

please read all comments and suggestions to improve your paper. Read carefully all comments and revise the paper. Also, check all journal guidelines.

Reviewers' comments:

Reviewer's Responses to Questions

**Comments to the Author**

1. If the authors have adequately addressed your comments raised in a previous round of review and you feel that this manuscript is now acceptable for publication, you may indicate that here to bypass the “Comments to the Author” section, enter your conflict of interest statement in the “Confidential to Editor” section, and submit your "Accept" recommendation.

Reviewer #1: All comments have been addressed

Reviewer #2: (No Response)

2. Is the manuscript technically sound, and do the data support the conclusions?

Reviewer #1: Yes

Reviewer #2: Partly

3. Has the statistical analysis been performed appropriately and rigorously?

Reviewer #1: Yes

Reviewer #2: N/A

4. Have the authors made all data underlying the findings in their manuscript fully available?

Reviewer #1: Yes

Reviewer #2: Yes

5. Is the manuscript presented in an intelligible fashion and written in standard English?

Reviewer #1: Yes

Reviewer #2: Yes

6. Review Comments to the Author

Reviewer #1: Thanks for fully addressing the concerns. It’s a nice paper, good work. I’m sure it will be useful.

Reviewer #2: Manuscript ID: PONE-D-24-33134R1

Title: missForestPredict – Missing data imputation for prediction settings

Dear Authors,

Thank you for your revised submission. After a careful and detailed review of your manuscript, I would like to share several concerns and points that require further attention and clarification. While your manuscript addresses an important topic, there are a number of methodological and conceptual issues that, in my opinion, weaken the current version and limit its clarity and practical utility.

1. Limited Justification for Convergence Criterion

Your proposed use of NMSE as a unified convergence criterion is not sufficiently justified. Specifically, applying the same NMSE thresholding logic to both continuous and categorical variables (via Brier Skill Score transformations) lacks nuance. The metric’s sensitivity to data imbalance—especially for categorical variables—is not discussed, and no rationale is provided for why the default convergence behavior (e.g., stopping when NMSE increases) is ideal across scenarios. This calls into question the robustness of the convergence mechanism.

Recommendation: Provide a more detailed rationale for choosing NMSE across variable types, include sensitivity analyses, and justify the default convergence thresholds through empirical or theoretical reasoning.

2. Scalability and Memory Use Concerns

The method stores imputation models for each variable and iteration, but the implications for scalability in high-dimensional datasets are not discussed. It is unclear how missForestPredict performs when applied to datasets with hundreds or thousands of variables (e.g., genomic or sensor data).

Recommendation: Include runtime and memory usage benchmarks on at least one high-dimensional dataset, or explicitly clarify the limitations and trade-offs of the current implementation in such settings.

3. Lack of Error Analysis

Although your results section reports predictive and imputation performance across scenarios, it fails to offer any qualitative or case-specific analysis of failures, misclassifications, or patterns of poor performance (e.g., in MNAR or MAR scenarios). This weakens interpretability and hinders practical guidance for users.

Recommendation: Include an error analysis section with case studies or diagnostic plots highlighting when and why the method fails or overfits (i.e., "over-imputes").

4. Insufficient Comparison to Contemporary Methods

While the study compares to several common imputation tools (e.g., mice, miceRanger, IterativeImputer), there is no mention or evaluation of modern deep learning or transformer-based imputation frameworks, or techniques that model missingness explicitly as informative.

Recommendation: Acknowledge the limitations of not including such baselines, or ideally, include one or two such recent methods for a fairer benchmark.

5. Ambiguity in Parameter Recommendations

Several user-controllable parameters (e.g., number of trees, max depth, p_obs, p_miss) are introduced but not thoroughly explored or explained in terms of their practical implications or impact on performance.

Recommendation: Add guidance, either empirically or heuristically derived, on how to set these parameters in various scenarios. This could be supported by a sensitivity analysis.

6. Poorly Structured Presentation of Results

The results section is overly long, dense, and at times repetitive. The placement of figures, especially for simulated data, precedes the more practically relevant real-data analysis. Important findings are buried in lengthy narrative without adequate summary or interpretation.

Recommendation: Restructure the results to highlight practical findings more clearly, streamline the narrative, and consider summarizing key patterns in performance using tables or concise bullet-point summaries.

7. Reproducibility and Workflow Transparency

While code is provided, there is no clear description of the workflow pipeline for reproducing the experiments (especially test-time imputation), nor a discussion on how the models generalize across datasets with varying missing data mechanisms.

Recommendation: Include a clear, schematic overview of the full workflow (perhaps in an appendix or figure), and discuss limitations in generalizing trained imputation models to datasets with differing missingness patterns.

In summary, although the manuscript presents an extension to an existing method, it currently lacks the depth and rigor necessary to fully support its contributions and claims. A substantial revision addressing the above concerns is required before this work can be considered for publication.

Sincerely,

7. PLOS authors have the option to publish the peer review history of their article (what does this mean?). If published, this will include your full peer review and any attached files.

Reviewer #1: **Yes: **David L Gibbs

Reviewer #2: No

---

## [Author Response · Author response to Decision Letter 2]

14 Aug 2025

Dear,

We have addressed all comments in the Response to Reviewers document.

Best regards,

Elena Albu (on behalf of all co-authors)

---

## [Decision Letter · Decision Letter 2]

2 Sep 2025

PONE-D-24-33134R2missForestPredict – Missing data imputation for prediction settingsPLOS ONE

Dear Dr. Van Calster,

Thank you for submitting your manuscript to PLOS ONE. After careful consideration, we feel that it has merit but does not fully meet PLOS ONE’s publication criteria as it currently stands. Therefore, we invite you to submit a revised version of the manuscript that addresses the points raised during the review process.

We look forward to receiving your revised manuscript.

Kind regards,

Leona Cilar Budler

Academic Editor

PLOS ONE

Journal Requirements:

1. the reviewer comments include a recommendation to cite specific previously published works, please review and evaluate these publications to determine whether they are relevant and should be cited. There is no requirement to cite these works unless the editor has indicated otherwise. 

Reviewers' comments:

Reviewer's Responses to Questions

**Comments to the Author**

1. If the authors have adequately addressed your comments raised in a previous round of review and you feel that this manuscript is now acceptable for publication, you may indicate that here to bypass the “Comments to the Author” section, enter your conflict of interest statement in the “Confidential to Editor” section, and submit your "Accept" recommendation.

Reviewer #1: All comments have been addressed

Reviewer #2: All comments have been addressed

2. Is the manuscript technically sound, and do the data support the conclusions?

Reviewer #1: Yes

Reviewer #2: Yes

3. Has the statistical analysis been performed appropriately and rigorously?

Reviewer #1: Yes

Reviewer #2: Yes

4. Have the authors made all data underlying the findings in their manuscript fully available?

Reviewer #1: Yes

Reviewer #2: Yes

5. Is the manuscript presented in an intelligible fashion and written in standard English?

Reviewer #1: Yes

Reviewer #2: Yes

6. Review Comments to the Author

Reviewer #1: Nice work, I the manuscript is great. I'm sure people will find it of use, especially in domains that more commonly have missing data.

Reviewer #2: Manuscript Title: missForestPredict – Missing data imputation for prediction settings

Manuscript Number: PONE-D-24-33134R2

Dear Authors,

I have carefully reviewed the revised manuscript and the corresponding “Response to Reviewers” document. The authors have made substantial and appropriate revisions to address the prior concerns raised by reviewers in earlier rounds. The paper now presents a clear, comprehensive, and well-motivated introduction of the missForestPredict algorithm, which meaningfully extends the widely used missForest imputation method to support prediction-time imputation — a feature of high practical value in numerous applied domains, particularly in clinical decision support, finance, and sensor-based analytics.

Remaining Suggestions (Minor, Stylistic/Grammatical):

- There are occasional typographical inconsistencies (e.g., spacing around mathematical operators, inconsistent use of “data are” vs. “data is”).

- Some sentences in the Methods section are long and could be split for improved readability, especially in the step-by-step algorithm description and convergence criterion explanation.

- Ensure uniform formatting of symbols and subscripts in equations (e.g., \(y^{(s)}_{obs}\) vs `y(s) obs`).

- Verify consistency in citation formatting according to journal style (e.g., spaces before references in square brackets).

- A quick language polishing pass may further enhance reader accessibility, particularly for non-specialist audiences.

Conclusion:

The manuscript presents a valuable, novel, and well-executed contribution to the literature on missing data imputation in prediction contexts. The work is methodologically sound, empirically well-validated, and will be of significant interest to applied researchers and practitioners.

Kind regards,

7. PLOS authors have the option to publish the peer review history of their article (what does this mean?). If published, this will include your full peer review and any attached files.

Reviewer #1: No

Reviewer #2: No

---

## [Author Response · Author response to Decision Letter 3]

18 Sep 2025

PONE-D-24-33134R2; RESPONSE TO REVIEWERS

REVIEWER #2

I have carefully reviewed the revised manuscript and the corresponding “Response to Reviewers” document. The authors have made substantial and appropriate revisions to address the prior concerns raised by reviewers in earlier rounds. The paper now presents a clear, comprehensive, and well-motivated introduction of the missForestPredict algorithm, which meaningfully extends the widely used missForest imputation method to support prediction-time imputation — a feature of high practical value in numerous applied domains, particularly in clinical decision support, finance, and sensor-based analytics.

Response: Thank you for your valuable input, and for these kind words.

Remaining Suggestions (Minor, Stylistic/Grammatical):

- There are occasional typographical inconsistencies (e.g., spacing around mathematical operators, inconsistent use of “data are” vs. “data is”).

Response: We have checked the text and fixed the few issues that we found.

- Some sentences in the Methods section are long and could be split for improved readability, especially in the step-by-step algorithm description and convergence criterion explanation.

Response: We have re-read this section, and we would respectfully like to keep it as is.

- Ensure uniform formatting of symbols and subscripts in equations (e.g., \(y^{(s)}_{obs}\) vs `y(s) obs`).

Response: We have checked the equations in the pdf file and we did not find inconsistencies. All equations appear correct to us.

- Verify consistency in citation formatting according to journal style (e.g., spaces before references in square brackets).

Response: We have checked in-text citation style and corrected a few formatting issues.

- A quick language polishing pass may further enhance reader accessibility, particularly for non-specialist audiences.

Response: Thank you for this comment! We made a small change to the abstract, and considerably enhanced the readability of the introduction and conclusion sections.

JOURNAL COMMENTS

1. the reviewer comments include a recommendation to cite specific previously published works, please review and evaluate these publications to determine whether they are relevant and should be cited. There is no requirement to cite these works unless the editor has indicated otherwise.

Response: unfortunately, in the decision e-mail we did not see any specific works the reviewers want us to cite. We are therefore unsure what this comment relates to.

Response: We updated reference 14 (from master thesis to journal publication). All other references appeared correct and up to date to us.

---

## [Decision Letter · Decision Letter 3]

23 Sep 2025

missForestPredict – Missing data imputation for prediction settings

PONE-D-24-33134R3

Dear Dr. Van Calster,

We’re pleased to inform you that your manuscript has been judged scientifically suitable for publication and will be formally accepted for publication once it meets all outstanding technical requirements.

Kind regards,

Leona Cilar Budler

Academic Editor

PLOS ONE

Additional Editor Comments (optional):

Reviewers' comments:

Reviewer's Responses to Questions

**Comments to the Author**

1. If the authors have adequately addressed your comments raised in a previous round of review and you feel that this manuscript is now acceptable for publication, you may indicate that here to bypass the “Comments to the Author” section, enter your conflict of interest statement in the “Confidential to Editor” section, and submit your "Accept" recommendation.

Reviewer #1: All comments have been addressed

Reviewer #2: All comments have been addressed

2. Is the manuscript technically sound, and do the data support the conclusions?

Reviewer #1: Yes

Reviewer #2: Yes

3. Has the statistical analysis been performed appropriately and rigorously?

Reviewer #1: Yes

Reviewer #2: Yes

4. Have the authors made all data underlying the findings in their manuscript fully available?

Reviewer #1: Yes

Reviewer #2: Yes

5. Is the manuscript presented in an intelligible fashion and written in standard English?

Reviewer #1: Yes

Reviewer #2: Yes

6. Review Comments to the Author

Reviewer #1: (No Response)

Reviewer #2: Dear Authors,

Thank you for carefully addressing the comments and suggestions raised during the review process. I have reviewed the revised manuscript, and I am pleased to note that all issues have been satisfactorily resolved. The final version is clear, well-prepared, and ready for publication.

Best regards,

7. PLOS authors have the option to publish the peer review history of their article (what does this mean?). If published, this will include your full peer review and any attached files.

Reviewer #1: No

Reviewer #2: No

---

## [Editor Report · Acceptance letter]

PONE-D-24-33134R3

PLOS ONE

Dear Dr. Van Calster,

I'm pleased to inform you that your manuscript has been deemed suitable for publication in PLOS ONE. Congratulations! Your manuscript is now being handed over to our production team.

Kind regards,

on behalf of

Dr. Leona Cilar Budler

Academic Editor

PLOS ONE